# HALL-E: Hierarchical Neural Codec Language Model for Minute-Long Zero-Shot Text-to-Speech Synthesis

**Yuto Nishimura**[1,2]**, Takumi Hirose**[2]**, Masanari Ohi**[2]**, Hideki Nakayama**[1]**, Nakamasa Inoue**[2]
[1] The University of Tokyo, [2] Institute of Science Tokyo
`yutonishimurav2@g.ecc.u-tokyo.ac.jp`

## Abstract

Recently, Text-to-speech (TTS) models based on large language models (LLMs) that translate natural language text into sequences of discrete audio tokens have gained great research attention, with advances in neural audio codec (NAC) models using residual vector quantization (RVQ). However, long-form speech synthesis remains a significant challenge due to the high frame rate, which increases the length of audio tokens and makes it difficult for autoregressive language models to generate audio tokens for even a minute of speech. To address this challenge, this paper introduces two novel post-training approaches: 1) Multi-Resolution Requantization (MReQ) and 2) HALL-E. MReQ is a framework to reduce the frame rate of pre-trained NAC models. Specifically, it incorporates multi-resolution residual vector quantization (MRVQ) module that hierarchically reorganizes discrete audio tokens through teacher-student distillation. HALL-E is an LLM-based TTS model designed to predict hierarchical tokens of MReQ. Specifically, it incorporates the technique of using MRVQ sub-modules and continues training from a pre-trained LLM-based TTS model. Furthermore, to promote TTS research, we create MinutesSpeech, a new benchmark dataset consisting of 40k hours of filtered speech data for training and evaluating speech synthesis ranging from 3s up to 180s. In experiments, we demonstrated the effectiveness of our approaches by applying our post-training framework to VALL-E. We achieved the frame rate down to as low as 8 Hz, enabling the stable minitue-long speech synthesis in a single inference step. Audio samples, dataset, codes and pre-trained models are available at `https://yutonishimura-v2.github.io/HALL-E_DEMO`.

## 1 Introduction

Recent advances in large language models (LLMs) have enabled us to model complex linguistic structures and patterns with unprecedented precision in natural language processing tasks (Brown et al., 2020; Chowdhery et al., 2023; Zeng et al., 2023). Motivated by these developments, LLM-based text-to-speech (TTS) models have gained research interest for their ability to model complex speech structures and patterns, enabling the synthesis of more natural-sounding speech in a zero-shot manner (Wang et al., 2023; Zhang et al., 2023b; Song et al., 2024; Han et al., 2024a; Chen et al., 2024a; Xin et al., 2024; Meng et al., 2024; Wang et al., 2024a). The core concept behind LLM-based TTS models is to translate natural language text into a sequence of audio tokens, typically using frozen natural audio codec (NAC) models that quantize audio signals into discrete tokens via vector quantization techniques (Zeghidour et al., 2021; Défossez et al., 2022; Kumar et al., 2023b; Zhang et al., 2024; Wu et al., 2023; Huang et al., 2023; Du et al., 2024; Yang et al., 2023a).

Since LLMs are becoming capable of capturing long context in text (Guo et al., 2022; Chen et al., 2024b; Han et al., 2024b), LLM-based TTS models are also expected to handle long context to synthesize speech over extended periods. However, this presents significant challenges, mainly because the length of audio tokens per second is typically large. More specifically, when using an autoregressive language model to predict audio tokens, as in the VALL-E architecture (Wang et al., 2023), the high frame rate[1] at the first layer of residual vector quantization (RVQ) (Zeghidour et al.,

---

[1]We refer to the number of audio tokens per second as the frame rate (Hz).

2021) in NAC models often becomes a major factor that hinders the synthesis of long speech. As recently discussed and investigated by Han et al. (2024a); Chen et al. (2024a), reducing the frame rate is essential, but not straightforward. Simply reducing the frame rate in NAC models results in degraded audio quality and incorrect phoneme pronunciation. Addressing this issue requires novel solutions that bridge NAC models and LLM-based TTS models.

In this paper, we propose a novel approach to tackle the challenge of minitue-long speech synthesis in LLM-based TTS models by introducing a hierarchical post-training framework that effectively manages the trade-off between reducing frame rate and producing high-quality speech. Our contributions are summarized as follows:

1) We propose **Multi-Resolution Requantization (MReQ)**, a post-training framework for hierarchically reorganizing pre-trained RVQ module to reduce the frame rate at lower quantization layers. MReQ incorporates a multi-resolution residual vector quantization (MRVQ) module into a pre-trained NAC model and continues training in a teacher-student distillation manner. This results in reducing the frame rate at the first quantization layer to 8 Hz.

2) We propose **HALL-E**, a hierarchical LLM-based TTS model designed to predict hierarchical tokens of MReQ. The AR model is trained using 8Hz tokens, while the NAR model is trained by using sub-modules in MRVQ, and continues training from a pre-trained LLM-based TTS model.

3) We introduce **MinutesSpeech**, a new benchmark dataset to promote TTS research, particularly for minute-long speech synthesis. The training set consists of 40k hours of automatically filtered and balanced speech data. The test set consists of 8 hours of speech data with transcriptions created by professional transcribers.

4) We thoroughly conducted experiments to provide best practices for managing the trade-off between reducing frame rate and producing high-quality speech, while demonstrating the effectiveness and efficiency of our approach. We open-source dataset, codes and pre-trained models along with audio samples at `https://yutonishimura-v2.github.io/HALL-E_DEMO`.

## 2 RELATED WORK

**Neural audio codec models.** NAC models produce discrete audio tokens by quantizing audio signals. SoundStream (Zeghidour et al., 2021) and Encodec (Défossez et al., 2022) are pioneering NAC models, which significantly improved compression efficiency over traditional audio codecs. Recent studies have proposed NAC models with a focus on maintaining performance in speech processing tasks. Examples include SpeechTokenizer (Zhang et al., 2024), Descript Audio Code (Kumar et al., 2023b), AcademiCodec (Yang et al., 2023a), AudioDec (Wu et al., 2023), RepCodec (Huang et al., 2023), and FunCodec (Du et al., 2024). Many studies have focused on reducing bps, while few have explored lowering and varying the frame rate.Dieleman et al. (2021) introduced the concept of time-varying audio codes. Défossez et al. (2024) achieved the lowest frame rate for a NAC model, reaching 12.5Hz by incorporating Transformer models and SpeechTokenizer. We report achieving an even lower frame rate of 8Hz, which is about 1.5 times shorter than it.

**Zero-shot TTS.** Zero-Shot TTS aims to synthesize speech from text in the voice of a target speaker using only a short reference audio segment from that speaker. Early models relied on speaker embeddings or speaker adaptation (Casanova et al., 2022; Arik et al., 2018; Chen et al., 2019), while recent studies have focused on LLM-based models that use NAC models in conjunction with LLMs. VALL-E (Wang et al., 2023) was the first LLM-based model, demonstrating impressive capabilities in zero-shot TTS tasks. Follow-up studies have explored various extensions such as VALL-E X for cross-lingual TTS (Zhang et al., 2023b), ELLA-V using Montreal forced aligner (Song et al., 2024), RALL-E using prosody features (Xin et al., 2024), VALL-E R using monotonic alignment (Han et al., 2024a), VALL-E 2 using grouped code modeling (Chen et al., 2024a), and MELLE using mel-spectrogram features (Meng et al., 2024). Prosody information can also be modeled by LLMs latent language mode Mega-TTS (Jiang et al., 2023) and Mega-TTS 2 (Jiang et al., 2024) introduced prosody LLMs to generate more natural prosody. Meanwhile, diffusion-based models (*e.g.,* NaturalSpeech2/3 (Shen et al., 2024; Ju et al., 2024) and Voicebox (Le et al., 2024)) and prompt-based models (*e.g.,* Prompt-TTS2 (Guo et al., 2023; Leng et al., 2024)) are also known to be effective to generate high-quality controllable speech. Beyond speech synthesis, several studies proposed audio generation models such as UniAudio (Yang et al., 2023b), Audiobox (Vyas et al., 2023). In contrast, this work explores post-training methods to reduce the frame rate of LLM-based models, aiming at minute-long speech synthesis given a pre-trained NAC model such as Encodec.

## 3 PRELIMINARIES

**Neural audio codec.** A NAC model typically consists of three components: an encoder $\text{Enc}(\cdot)$, a vector quantizer $\text{VQ}(\cdot)$, and a decoder $\text{Dec}(\cdot)$. The quantizer is the core component that produces discrete audio tokens. This work assumes that an RVQ module (Zeghidour et al., 2021) is used as the quantizer, which is defined as follows:

$$\boldsymbol{z}_l = \text{VQ}_l(\boldsymbol{x}_{l-1}), \quad (1) \qquad \boldsymbol{x}_l = \boldsymbol{x}_{l-1} - \tilde{\boldsymbol{z}}_l, \quad (2) \qquad \boldsymbol{h} = \sum_{l=1}^{L} \tilde{\boldsymbol{z}}_l, \quad (3)$$

where $\boldsymbol{x}_0 = \text{Enc}(\boldsymbol{x}_{\text{in}}) \in \mathbb{R}^{d \times n}$ is the encoder output for the input audio $\boldsymbol{x}_{\text{in}}$, $d$ is the latent dimension, $n$ is the sequence length, $\text{VQ}_l$ is a vector quantizer, $\boldsymbol{z}_l \in \mathbb{N}^n$ is a discrete token sequence, $\tilde{\boldsymbol{z}}_l = \text{Emb}_l(\boldsymbol{z}_l) \in \mathbb{R}^{d \times n}$ is a sequence of embeddings corresponding to $\boldsymbol{z}_l$ obtained through a learnable embedding layer $\text{Emb}_l(\cdot)$[2], $l \in \{1, 2, \cdots, L\}$ is the layer index, and $L$ is the number of layers. The output $\boldsymbol{h} \in \mathbb{R}^{d \times n}$ is then fed into the decoder to reconstruct the input audio as $\boldsymbol{y} = \text{Dec}(\boldsymbol{h})$.

**LLM-based TTS.** An LLM-based TTS typically consists of two decoder-only language models: an autoregressive (AR) model $T_{\text{ar}}$ and a non-autoregressive (NAR) model $T_{\text{nar}}$ (Wang et al., 2023). The speech synthesis procedure is given by the following equations:

$$\hat{\boldsymbol{z}}_1 = T_{\text{ar}}(\boldsymbol{t}, \boldsymbol{z}_1^{\text{pr}}), (4) \ \ \hat{\boldsymbol{z}}_{l+1} = T_{\text{nar}}(\boldsymbol{t}, \boldsymbol{h}_L^{\text{pr}}, \hat{\boldsymbol{h}}_l, l), (5) \ \ \hat{\boldsymbol{h}}_l = \sum_{l'=1}^{l} \hat{\tilde{\boldsymbol{z}}}_{l'}, (6) \ \ \hat{\boldsymbol{y}} = \text{Dec}([\boldsymbol{h}_L^{\text{pr}}, \hat{\boldsymbol{h}}_L]), (7)$$

where $[\boldsymbol{h}_L^{\text{pr}}, \hat{\boldsymbol{h}}_L]$ denotes the concatenation of these two matrices along the time axis. In Eq. (4), $T_{\text{ar}}$ generates an audio token sequence $\hat{\boldsymbol{z}}_1 \in \mathbb{N}^{n'}$ corresponding to the first layer of RVQ given two inputs: a text prompt $\boldsymbol{t}$ and an audio prompt $\boldsymbol{z}_1^{\text{pr}} = \text{VQ}_1(\text{Enc}(\boldsymbol{x}_{\text{pr}}))$ extracted from an audio input $\boldsymbol{x}_{\text{pr}}$. In Eq. (5), $T_{\text{nar}}$ iteratively generates token sequences $\hat{\boldsymbol{z}}_{l+1} \in \mathbb{N}^{n'}$ from the accumulated hidden features $\hat{\boldsymbol{h}}_l$ in Eq. (6) and the audio prompt's hidden features $\boldsymbol{h}_L^{\text{pr}}$. Finally, in Eq. (7), speech $\hat{\boldsymbol{y}}$ is generated. Note that $\text{Enc}$, $\text{VQ}_1$, and $\text{Dec}$ are from a frozen NAC model.

**Preliminary experiments.** LLM-based TTS models have a predefined context window size and are typically trained with speech data ranging from several seconds to several tens of seconds. To generate long speech segments, a straightforward approach is to reduce the frame rate in the NAC model. However, reducing the frame rate below 48 Hz significantly decreases speech reconstruction performance as shown in Figure 1, where we evaluated the performance of Encodec in terms of the estimated Perceptual Evaluation of Speech Quality (PESQ) scores (Kumar et al., 2023a) and word error rates (WERs) as functions of frame rates. Specifically, it is confirmed that training becomes entirely difficult at 8Hz. Therefore, in this study, we propose a NAC model that works even at an 8Hz, demonstrating a significant improvement over existing limitations.

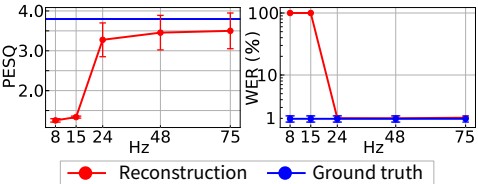

Figure 1: Preliminary experiments on varying the frame rate of Encodec.

## 4 MREQ: MULTI-RESOLUTION REQUANTIZATION

This section introduces MReQ, a post-training framework for hierarchically reorganizing a pre-trained RVQ module to reduce the frame rate. Specifically, MReQ incorporates a multi-resolution residual vector quantization (MRVQ) module to a pre-trained NAC model as shown in Figure 2, and continues training the NAC model in a teacher-student distillation manner. For a pre-trained 48Hz Encodec model, MReQ reduces the frame rate at the first quantization layer to 8 Hz. This enables LLM-based TTS models to handle longer contexts.

### 4.1 ARCHITECTURE

The MRVQ module is a nested structure of RVQ. Specifically, it consists of a residual structure composed of multiple low frame-rate residual vector quantization (LRVQ) blocks, each of which is itself a residual structure operating at a different frame rate. The definition is given as follows.

---

[2] In this paper, the tilde symbol ($\tilde{\ }$) denotes the embeddings corresponding to a token sequence.

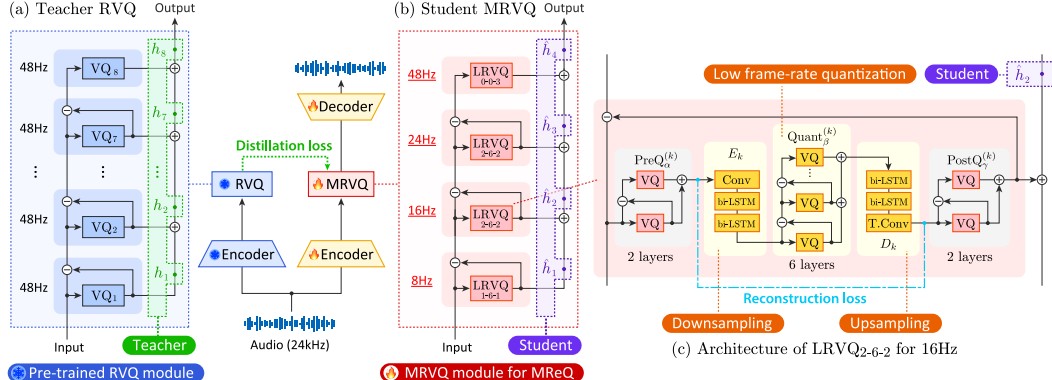

Figure 2: **MReQ post-training based on teacher-student distillation.** (a) Pre-trained RVQ module is used to extract teacher embeddings $\boldsymbol{h}_t$. (b) MRVQ module consists of multiple LRVQ blocks and learns to reduce the frame rates. Student embeddings $\boldsymbol{h}_s$ are extracted. (c) Each LRVQ block consists of a pre-quantizer $\mathrm{PreQ}$, a sub-encoder $E$, a main quantizer $\mathrm{Quant}$, a sub-decoder $D$, and a post-quantizer $\mathrm{PostQ}$ to reduce frame rate from $s_0$ to $s_k$.

**Definition 1 (MRVQ module).** Let $\boldsymbol{x}_0 \in \mathbb{R}^{d \times n_0}$ be an encoder output, where $d$ is the latent dimension and $n_0 = Ts_0$ is the sequence length depending on the time length $T$ (sec) and the frame rate $s_0$ (Hz). The MRVQ module is defined as follows:

$$\boldsymbol{c}_k = \mathrm{LRVQ}_{\alpha\text{-}\beta\text{-}\gamma}^{(k)}(\boldsymbol{x}_{k-1}), \quad (8) \qquad \boldsymbol{x}_k = \boldsymbol{x}_{k-1} - \tilde{\boldsymbol{c}}_k, \qquad (9) \qquad \boldsymbol{h} = \sum_{k=1}^{K} \tilde{\boldsymbol{c}}_k, \qquad (10)$$

where $\mathrm{LRVQ}_{\alpha\text{-}\beta\text{-}\gamma}^{(k)}$ is an LRVQ block, $K$ is the number of blocks, $\alpha\text{-}\beta\text{-}\gamma$ is a triplet of hyperparameters to determine the block structure.

**Definition 2 (LRVQ block).** Each LRVQ block $\mathrm{LRVQ}_{\alpha\text{-}\beta\text{-}\gamma}^{(k)}$ consists of five components: a pre-quantizer $\mathrm{PreQ}_\alpha^{(k)}$, a sub-encoder $E_k$ for down sampling, a main quantizer $\mathrm{Quant}_\beta^{(k)}$, a sub-decoder $D_k$ for upsampling, and a post-quantizer $\mathrm{PostQ}_\gamma^{(k)}$. The quantization procedure is given by

$$\boldsymbol{a}_k = \mathrm{PreQ}_\alpha^{(k)}(\boldsymbol{x}_{k-1}), \; (11) \qquad \boldsymbol{b}_k = \mathrm{Quant}_\beta^{(k)}(E_k(\tilde{\boldsymbol{a}}_k)), \; (12) \qquad \boldsymbol{c}_k = \mathrm{PostQ}_\gamma^{(k)}(D_k(\tilde{\boldsymbol{b}}_k)), \; (13)$$

where $\boldsymbol{a}_k, \boldsymbol{b}_k, \boldsymbol{c}_k$ are token sequences. The three quantizers $\mathrm{PreQ}_\alpha^{(k)}, \mathrm{Quant}_\beta^{(k)}$ and $\mathrm{PostQ}_\gamma^{(k)}$ are implemented using RVQ with $\alpha$, $\beta$, and $\gamma$ layers, respectively. Note that $\boldsymbol{b}_k \in \mathbb{N}^{\beta \times n_k}$ is the token sequence representing audio in a low frame rate. Its length is given by $n_k = Ts_k$, where $s_k$ is the frame rate satisfying $s_1 < s_2 < \cdots < s_K$ and $s_K = s_0$. The other two sequences $\boldsymbol{a}_k$ and $\boldsymbol{c}_k$ are used only for facilitating training of NAR models used in LLM-based TTS models.

**Implementation details.** Figure 2b shows the MRVQ module applied to the Encodec model, where the frame rate is reduced from $s_0 = 48$ Hz to $s_1 = 8$ Hz using 4 LRVQ blocks. Figure 2c shows the LRVQ block. Each sub-encoder $E_k$ consists of a convolution layer followed by two bi-LSTM layers, which reduces the frame rate from $s_0$ to $s_k$. Each sub-decoder $D_k$ consists of two bi-LSTM layers followed by a transposed convolution layer, which is symmetric to $E_k$. Table 1 lists frame rates $s_k$, hyperparameter triplets $\alpha\text{-}\beta\text{-}\gamma$, and strides for the convolution and transposed convolution layers. For $k = 4$, only the pre-quantize is used, and the other components are replaced with identical functions, reducing Eqs. (12, 13) to $\boldsymbol{b}_4 = \boldsymbol{a}_4$ and $\boldsymbol{c}_4 = \boldsymbol{b}_4$, respectively.

Table 1: Implementation details

| $k$ | $s_k$ | $\alpha\text{-}\beta\text{-}\gamma$ | Stride |
|---|---|---|---|
| 1 | 8 | 1-6-1 | 6 |
| 2 | 16 | 2-6-2 | 3 |
| 3 | 24 | 2-4-2 | 2 |
| 4 | 48 | 3-0-0 | - |

### 4.2 POST-TRAINING WITH TEACHER-STUDENT DISTILLATION

Training NAC models with the MRVQ module is challenging because quantization layers with lower frame rates are prone to be ignored. To address this, we introduce a post-training technique based on teacher-student distillation, where a NAC model pre-trained with a high frame rate serves as the teacher model. As shown in Figure 2a, teacher embeddings (in green) are extracted from the frozen RVQ module, while student embeddings (in purple) are extracted from the MRVQ module. We then minimize the feature-level distillation (FLD) loss and the hidden-state reconstruction (HSR) loss.

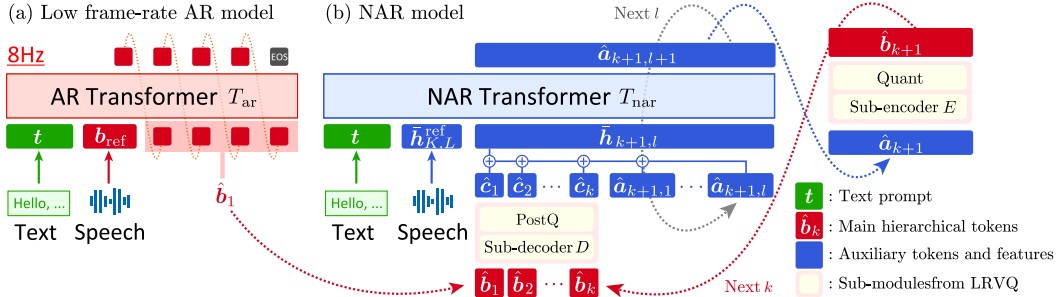

Figure 3: **HALL-E architecture.** (a) AR model generates a low frame-rate token sequence $\hat{\boldsymbol{b}}_1$. (b) NAR model predicts $\hat{\boldsymbol{b}}_{k+1}$ from $\hat{\boldsymbol{b}}_k$ iteratively by utilizing frozen sub-modules of MRVQ.

**FLD loss.** The FLD loss is an MAE loss between teacher and student embeddings, defined as

$$L_{\text{FLD}} = \sum_{(s,t) \in \mathcal{P}} \lambda_{(s,t)}^{\text{FLD}} \|\hat{\boldsymbol{h}}_s - \boldsymbol{h}_t\|_1, \quad (14) \qquad \hat{\boldsymbol{h}}_s = \sum_{k=1}^{s} \tilde{\boldsymbol{c}}_k, \quad (15) \qquad \boldsymbol{h}_t = \sum_{l=1}^{t} \tilde{\boldsymbol{z}}_l, \quad (16)$$

where $\boldsymbol{h}_s$ is a student embedding, $\boldsymbol{h}_t$ is a teacher embedding, $\mathcal{P}$ is a set of student-teacher index pairs, and $\lambda_{(s,t)}^{\text{FLD}}$ is a weight coefficient. We use $\mathcal{P} = \{(1,1),(2,3),(3,5),(4,8)\}$ for RVQ with eight layers and MRVQ with four LRVQ blocks. Note that $\tilde{\boldsymbol{c}}_k$ and $\tilde{\boldsymbol{z}}_l$ are obtained from Eqs. (8) and Eqs. (1), respectively. The student-teacher pairs in $\mathcal{P}$ are determined so that the cumulative number of student's post-quantization layers matches that of the teacher's quantization layers.

**HSR loss.** The HSR loss is introduced to further facilitate training of each LRVQ block:

$$L_{\text{HSR}} = \sum_{k=1}^{K} \lambda_k^{\text{HSR}} \|\tilde{\boldsymbol{a}}_k - D_k(\tilde{\boldsymbol{b}}_k)\|_1 \qquad (17)$$

where $\tilde{\boldsymbol{a}}_k, \tilde{\boldsymbol{b}}_k$ are from Eqs. (11, 12), and $\lambda_k^{\text{HSR}}$ is a weight coefficient.

**Total loss.** The total loss is given by $L_{\text{total}} = L_{\text{NAC}} + L_{\text{FLD}} + L_{\text{HSR}}$, where $L_{\text{NAC}}$ is the loss used to train the NAC model. We continue to train the encoder and decoder with the MRVQ module using a copied weight from the NAC model, which is used as a teacher model. Compared to training from scratch, this allows for more efficient and stable convergence.

**Discussion.** Our post-training approach is designed to be independent of the encoder-decoder architecture of the original NAC model, as we only assumed the use of RVQ. Consequently, by utilizing state-of-the-art NAC models such as SpeechTokenizer (Zhang et al., 2024) instead of Encodec (Défossez et al., 2022), it is possible to achieve higher performance at a lower frame rate.

## 5  HIERARCHICAL NEURAL CODEC LANGUAGE MODEL

This section introduces HALL-E, a hierarchical LLM-based TTS model that learns to generate hierarchical tokens. Inspired by VALL-E (Wang et al., 2023), HALL-E consists of a pair of language models: an AR model and an NAR model. There are two key differences compared to VALL-E. First, our AR model handles low frame-rate sequences, as low as 8 Hz in our experiments, enabling stable generation of audio tokens for longer speech segments. Second, we incorporate frozen sub-modules obtained by decomposing the MRVQ module into the token prediction process using the NAR model. This integration facilitates training and results in high-quality speech synthesis. The whole inference process is illustrated in Figure 3.

**AR model.** Given a text prompt $\boldsymbol{t}$ and an audio $\boldsymbol{x}_{\text{pr}}$ as a prompt, the AR model predicts a low frame-rate token sequence $\hat{\boldsymbol{b}}_1$ corresponding to $\boldsymbol{t}$. This step is formulated as $\hat{\boldsymbol{b}}_1 = T_{\text{ar}}(\boldsymbol{t}, \boldsymbol{b}_1^{\text{pr}})$, similar to Eq. (4) for VALL-E, where $T_{\text{ar}}$ is an AR transformer decoder and $\boldsymbol{b}_1^{\text{pr}}$ is the audio prompt obtained by the first LRVQ block.

**NAR model.** Given a token sequence $\hat{\boldsymbol{b}}_k$, the NAR model predicts the next token sequence $\hat{\boldsymbol{b}}_{k+1}$ iteratively. Specifically, $\hat{\boldsymbol{b}}_{k+1}$ is predicted in three steps by utilizing frozen sub-modules obtained from the MRVQ module. First, the sub-decoder and the post-quantizer are applied to $\hat{\boldsymbol{b}}_k$ to ob-

tain $\hat{\boldsymbol{c}}_k = \mathrm{PostQ}_\gamma^{(k)}(D_k(\hat{\bar{\boldsymbol{b}}}_k))$. Second, an NAR transformer decoder $T_{\mathrm{nar}}$ is employed to predict $\hat{\boldsymbol{a}}_{k+1}$ from $\hat{\boldsymbol{c}}_k$. Because MRVQ has a nested structure, this step requires applying the decoder $\alpha_{k+1}$ times, where $\alpha_{k+1}$ is the number of layers in the pre-quantizer at the block $k+1$. Specifically, for the layer $l+1$ in block $k+1$, denoted as $\hat{\boldsymbol{a}}_{k+1,l+1}$, is computed $\alpha_{k+1}$ times using the following equation: $\hat{\boldsymbol{a}}_{k+1,l+1} = T_{\mathrm{nar}}(\boldsymbol{t}, \bar{\boldsymbol{h}}_{K,L}^{\mathrm{pr}}, \bar{\boldsymbol{h}}_{k+1,l}, \ell_{k+1,l})$ similar to Eq. (5), where $\bar{\boldsymbol{h}}_{k+1,l} = \sum_{k'=1}^{k} \hat{\bar{\boldsymbol{c}}}_{k'} + \sum_{l'=1}^{l} \hat{\bar{\boldsymbol{a}}}_{k+1,l'}$ is the accumulated feature and $\bar{\boldsymbol{h}}_{K,L}^{\mathrm{pr}}$ is the one for prompt. The layer id $\ell_{k+1,l} = \sum_{k'=1}^{k} \alpha_k + l$ is the cumulative sum of the number of pre-quantization layers. Finally, $\hat{\boldsymbol{b}}_{k+1}$ is obtained by applying the sub-encoder and the main quantizer to $\hat{\boldsymbol{a}}_{k+1}$ as $\hat{\boldsymbol{b}}_{k+1} = \mathrm{Quant}^{(k+1)}(E_{k+1}(\hat{\bar{\boldsymbol{a}}}_{k+1}))$. This process enables the NAR model to effectively generate high-fidelity token sequences.

**Training.** The AR model is trained using a cross-entropy loss $L_{\mathrm{CE}}(\hat{\boldsymbol{b}}_1, \boldsymbol{b}_1)$ measured between a prediction $\hat{\boldsymbol{b}}_1$ and the corresponding ground truth $\boldsymbol{b}_1$ obtained from training speech data. The delay pattern used in MusicGen (Copet et al., 2023a) is also applied. The NAR model is also trained using a cross-entropy loss $L_{\mathrm{CE}}(\hat{\boldsymbol{a}}_{k+1,l+1}, \boldsymbol{a}_{k+1,l+1})$, where $\hat{\boldsymbol{a}}_{k+1,l+1}$ is a prediction and $\boldsymbol{a}_{k+1,l+1}$ is the ground truth. HALL-E is also post-trained given a pre-trained LLM-based TTS model.

**Discussion.** This work focuses on post-training for reducing frame rate, but further exploration in token merging and grouping structures (Han et al., 2024a; Chen et al., 2024a) would also be interesting in future research. Another architecture design involves incorporating learnable upsampling layers into the NAR model to predict $\hat{\boldsymbol{b}}_{k+1}$ from $\hat{\boldsymbol{b}}_k$. However, handling different frame rates with a single NAR transformer is challenging. Additionally, a sub-encoder and sub-decoder are placed between $\hat{\boldsymbol{b}}_{k+1}$ and $\hat{\boldsymbol{b}}_k$, making the relationships between them more complex compared to the RVQ-based approach. To mitigate this, we employ the MRVQ sub-modules like above, which simplify the relationships to resemble the previous method, thereby improving the efficiency of the training.

## 6 MINUTESSPEECH BENCHMARK DATASET

This section introduces MinutesSpeech, a benchmark dataset for minutes-long TTS synthesis. Unlike previous datasets such as LibriSpeech, which are primarily designed for automatic speech recognition, our dataset is curated to advance TTS research from the following three perspectives.

**1) Benchmarking Minutes-long TTS Synthesis.** We provide two subsets for benchmarking: MinutesSpeech-90s and MinutesSpeech-180s, consisting of speech segments ranging from 3 seconds to 90 seconds and 3 seconds to 180 seconds, respectively. All test audio files are under Creative Commons licenses. For each speech segment, we provide transcriptions created by two professional native transcribers. Since LLM-based TTS models have been typically evaluated using audio segments ranging from 4 to 10 seconds from LibriSpeech in previous studies, our dataset promotes research on longer speech synthesis by providing substantially longer segments.

**2) Balanced Audio Length for Training.** We also provide a training set consisting of 40,000 hours of audio data. To successfully train TTS models capable of stably generating audio ranging from a few seconds to over a minute in a single inference, we carefully designed the distribution of audio lengths in the training dataset, covering durations from 3 seconds up to 180 seconds. Automatically generated transcriptions are provided for this subset to facilitate large-scale training.

**3) Variety of Speaking Styles.** To encompass a variety of conversational speech styles, we curated data from podcasts. Compared to the audiobook speech in LibriSpeech, synthesizing conversational speech is more challenging due to its spontaneous nature. We believe this is an important research direction that can help bridge the gap between LLMs and LLM-based TTS models.

**Data Curation Pipeline.** To create the training set, we first curated 296,464 podcast files, ranging from 10 to 90 minutes, totaling about 186k hours. We evaluated these audio files using the PESQ score estimated using TorchAudio-Squim (Kumar et al., 2023a). The audio was segmented every 30 seconds, and the score was calculated for each segment. We then computed the mean and standard deviation for each audio file, retaining only those with a mean score higher than 2.5 and a standard deviation lower than 0.6. As a result, approximately 25% of the data remained. We then applied automatic speech recognition and speaker diarization to extract speech segments, each associated with a single speaker. Specifically, Whisper distil Large

Table 2: Dataset statistics calculated on audio segments longer than 3s. † marks the results at 8Hz.

| Dataset | Split | Clips | Audio duration | | Words | Max Token length | | |
|---|---|---|---|---|---|---|---|---|
| | | | Average (sec) | Total (hours) | | Audio | Text | Sum |
| LibriSpeech | train-clean-100 | 27949 | 12.90 $\pm 3.29$ | 100.18 | 986k | 1176 | 240 | 1416 |
| LibriSpeech | test-35s | 1937 | 8.29 $\pm 5.10$ | 4.46 | 43k | 1680 | 340 | 2020 |
| MinutesSpeech | train-28s | 7385k | 19.11 $\pm 7.31$ | 39.21k | 407501k | 1344 | 314 | 1658 |
| MinutesSpeech | train-90s | 4181k | 34.37 $\pm 28.32$ | 39.92k | 407689k | 720† | 933 | 1655 |
| MinutesSpeech | train-54s | 4984k | 28.82 $\pm 17.45$ | 39.91k | 409261k | 2592 | 579 | 3171 |
| MinutesSpeech | train-180s | 3762k | 37.69 $\pm 41.94$ | 39.39k | 401816k | 1440† | 1733 | 3173 |
| MinutesSpeech | test-90s | 685 | 45.28 $\pm 31.35$ | 8.61 | 83k | 720† | 925 | 1645 |
| MinutesSpeech | test-180s | 536 | 56.76 $\pm 55.24$ | 8.45 | 81k | 1440† | 1715 | 3155 |

v3[3] was used for automatic transcriptions, and Pyannote[4] was employed for speaker diarization. We then segmented each audio file into segments ranging from 3 to 90 or 180 seconds. Lastly, we removed a certain proportion of utterances with text lengths that were either too long or too short, based on their respective distributions, resulting in a final dataset of approximately 40k hours. For the test set, we manually collected audio files under Creative Commons licenses and asked professional transcribers to select high-quality audio and create accurate transcriptions. Note that filtering based on the maximum and minimum text lengths used in the training set was also applied to each test set.

**Dataset Statistics.** Figure 4 shows the distributions of speech durations and the number of words per segment in the test sets. As shown, our dataset provides a diverse range of speech lengths, facilitating the development and evaluation of TTS models capable of handling varying durations and linguistic content.

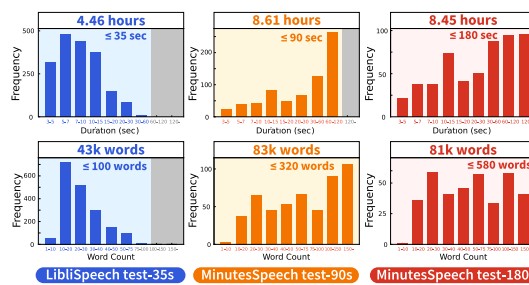

Figure 4: Duration and word count distributions.

# 7 EXPERIMENTS

## 7.1 EXPERIMENTAL SETUP

**Baseline models.** As a competitive comparison, we chose VALL-E (Wang et al., 2023) using Encodec (Défossez et al., 2022) at 48Hz as the baseline model, which is already lower than 75Hz used in previous studies, based on our preliminary experiments (see Figure 1). We also demonstrate the effectiveness of our approach on SpeechTokenizer (Zhang et al., 2024).

**Datasets.** For training, we used the MinutesSpeech training set, specifically train-90s and -180s for HALL-E, and train-28s, -54s, -90s, and -180s for VALL-E. The train-28s and -54s are designed for 48Hz to match the token length in train-90s and -180s at 8Hz (see Table 2). For evaluation, MinutesSpeech test-90s, -180s, and LibriSpeech test_clean set are used. The minimum audio length was set to 4s, while the maximum audio lengths were set to 90s, 180s, and 35s, respectively. The audio length for prompt was consistently set to 3s.

**Evaluation metrics.** Two evaluation metrics are used for speech reconstruction experiments: WER and PESQ. WER is calculated using the conformer-transducer[5]. Zero-shot TTS experiments are conducted in the continual setting (Wang et al., 2023). Seven evaluation metrics are used: WER, speaker similarity (SIM), DNSMOS using ITU-T P.808 (ITU, 2018)[6], UTMOS (Saeki et al., 2022), Wasserstein distance (WD) with respect to the duration distribution, subjective evaluation of naturalness (QMOS), and subjective evaluation of speaker similarity (SMOS). SIM is calculated using WavLM-TDNN [7]. WD is calculated between the duration distributions of the generated speech and the ground truth speech. For QMOS and SMOS, 40 utterances were randomly selected from each

---

[3] https://huggingface.co/distil-whisper/distil-large-v3
[4] https://huggingface.co/pyannote/speaker-diarization-3.1
[5] https://huggingface.co/nvidia/stt_en_conformer_transducer_xlarge
[6] https://github.com/microsoft/DNS-Challenge/tree/master/DNSMOS
[7] https://github.com/microsoft/UniSpeech/tree/main/downstreams/speaker_verification

test set, and three native English speakers rated their naturalness on a scale from 1 to 5. We then calculated the mean and confidence intervals.

**Training details.** Encodec and SpeechTokenizer were pre-trained using the Adam optimizer for 100k iters with a batch size of 704 for Encodec and 674 for SpeechTokenizer on four H100 GPUs, and a learning rate of $9 \times 10^{-4}$. MReQ post-training was performed on a single H100 GPU for 160k iters with a batch size of 160 and a learning rate of $3 \times 10^{-4}$. VALL-E was trained using the AdamW optimizer for 100k iters on four H100 GPUs. To fully utilize GPU memory, the batch size was adjusted based on the audio length of the training samples. A cosine annealing learning rate schedule was employed with an initial learning rate of $1 \times 10^{-4}$. HALL-E was trained using the same settings, where VALL-E is used as a pre-trained model. More details are provided in Appendix A.

## 7.2 SPEECH RECONSTRUCTION

Table 3 shows the speech reconstruction performance of Encodec and SpeechTokenizer before and after applying MReQ. MReQ maintains clarity and naturalness in terms of WER, STOI, SIM and UTMOS, while significantly reducing the frame rate at the first quantization layer to as low as 8 Hz.

Table 3: Speech reconstruction on LibriSpeech.

| NAC (frame rate, Hz) | WER↓ | PESQ↑ | STOI↑ | SIM↑ | UTMOS↑ |
|---|---|---|---|---|---|
| GT | 1.96 | $4.64_{\pm 0.00}$ | $1.00_{\pm 0.00}$ | 1.00 | $4.05_{\pm 0.33}$ |
| Encodec (8) | 100.0 | $1.22_{\pm 0.08}$ | $0.32_{\pm 0.05}$ | 0.52 | $1.33_{\pm 0.00}$ |
| Encodec (24) | 2.01 | $3.68_{\pm 0.17}$ | $0.94_{\pm 0.03}$ | 0.90 | $3.82_{\pm 0.37}$ |
| Encodec (48) | 2.00 | $4.02_{\pm 0.13}$ | $0.96_{\pm 0.02}$ | 0.94 | $3.86_{\pm 0.36}$ |
| +MReQ (8,16,24,48) | 2.02 | $3.89_{\pm 0.15}$ | $0.95_{\pm 0.02}$ | 0.92 | $3.89_{\pm 0.36}$ |
| SpeechTokenizer (48) | 2.00 | $4.10_{\pm 0.11}$ | $0.96_{\pm 0.02}$ | 0.95 | $3.97_{\pm 0.33}$ |
| +MReQ (8,16,24,48) | 1.99 | $3.96_{\pm 0.14}$ | $0.95_{\pm 0.02}$ | 0.93 | $4.01_{\pm 0.32}$ |

## 7.3 ZERO-SHOT SPEECH SYNTHESIS

**MinutesSpeech evaluation.** Table 4 compares performance of zero-shot TTS synthesis on the MinutesSpeech test sets. Overall, HALL-E significantly outperformed the VALL-E, achieving comparable or even better WER and DNSMOS scores than the ground truth audio. This demonstrates the effectiveness of our approach. VALL-E resulted in a high WER and a low QMOS score when trained on train-28s because it cannot generate audio longer than 28 seconds. When trained on train-90s, the WER decreased but still remained significantly higher than that of ground truth. This is because training an AR model with a high frame rate is unstable, indicating that simply changing the training data is not sufficient for successful training. Our approach addressed this issue by lowering the frame rate via MReQ. In terms of SIM, VALL-E outperformed HALL-E because lowering the frame rate sacrifices acoustic information. However, HALL-E surpassed VALL-E in SMOS, as it generates the speaker's important style element, duration, much more naturally than VALL-E (see Figure 6).

**LibriSpeech evaluation.** Table 5 shows results on LibriSpeech for short speech synthesis up to 35 seconds. As shown, VALL-E's WER increased as the training audio length increases. In contrast, HALL-E achieved a WER lower than 5% even with train-180s. The gap in WER between the model and the ground truth is due to the difference between LibriSpeech, which primarily contains read speech, and MinutesSpeech, which includes a significant amount of spontaneous speech. Balancing speech synthesis of both read and spontaneous speech remains a challenge for future work.

**NAC models.** Table 6 compares the results obtained by Encodec and SpeechTokenizer. As shown, all metrics except SIM improve with the use of SpeechTokenizer. Since SpeechTokenizer aims to preserve more linguistic information in the first VQ layer, it is likely that this aligns well with maintaining linguistic information better than acoustic information when the frame rate is reduced. These results suggest the potential for further performance enhancement by employing more advanced NAC models in the future.

**Computational efficiency.** Table 5 compares the real-time factor (RTF) of VALL-E and HALL-E, measured on an RTX 4090 GPU using the 4s to 10s segments from LibriSpeech. The results show that HALL-E generates audio approximately **3.4 times faster** than VALL-E. In LLM-based TTS, the computational bottleneck typically lies in the AR model. Significant speed improvements were achieved by HALL-E because it reduces the number of tokens required for generation by a factor of six. This enhancement shows promising potential not only for LLM-based TTS but also for various applications, such as spoken language modeling, as part of future work.

Table 4: Zero-shot TTS performance on MinutesSpeech test sets. Best results are marked in bold.

| | TTS model | NAC model | Training | WER↓ | SIM↑ | WD↓ | DNSMOS↑ | UTMOS↑ | QMOS↑ | SMOS↑ |
|---|---|---|---|---|---|---|---|---|---|---|
| test-90s | GT | – | – | 10.30 | – | – | $3.79 \pm 0.24$ | $3.28 \pm 0.51$ | $3.83 \pm 0.30$ | - |
| | VALL-E | Encodec | train-28s | 39.77 | **0.726** | 23.62 | $3.84 \pm 0.19$ | $3.61 \pm 0.48$ | $2.29 \pm 0.32$ | $2.04 \pm 0.25$ |
| | VALL-E | Encodec | train-90s | 16.14 | 0.712 | **2.68** | $3.87 \pm 0.17$ | $3.58 \pm 0.60$ | $2.48 \pm 0.28$ | $2.36 \pm 0.26$ |
| | HALL-E | MReQ-Encodec | train-90s | **9.79** | 0.685 | 4.00 | $\mathbf{3.91 \pm 0.21}$ | $\mathbf{3.74 \pm 0.37}$ | $\mathbf{3.35 \pm 0.26}$ | $\mathbf{3.15 \pm 0.26}$ |
| test-180s | GT | – | – | 10.20 | – | – | $3.78 \pm 0.25$ | $3.25 \pm 0.50$ | $4.26 \pm 0.22$ | - |
| | VALL-E | Encodec | train-54s | 36.52 | **0.706** | 25.66 | $3.55 \pm 0.53$ | $3.15 \pm 0.97$ | $1.68 \pm 0.25$ | $1.70 \pm 0.26$ |
| | VALL-E | Encodec | train-180s | 21.71 | 0.702 | 12.52 | $3.76 \pm 0.30$ | $3.55 \pm 0.62$ | $2.11 \pm 0.26$ | $2.15 \pm 0.26$ |
| | HALL-E | MReQ-Encodec | train-180s | **10.53** | 0.660 | **5.79** | $\mathbf{3.91 \pm 0.19}$ | $\mathbf{3.74 \pm 0.37}$ | $\mathbf{3.38 \pm 0.25}$ | $\mathbf{3.31 \pm 0.23}$ |

Table 5: Zero-shot TTS performance on LibriSpeech test_clean set. Best results in the same group (1 or 2) are marked in bold. QMOS and SMOS were conducted in the same group.

| TTS model | NAC model | Training | WER↓ | SIM↑ | WD↓ | DNSMOS↑ | UTMOS↑ | QMOS↑ | SMOS↑ |
|---|---|---|---|---|---|---|---|---|---|
| GT[1] | - | - | 1.74 | - | - | $3.84 \pm 0.19$ | $4.07 \pm 0.31$ | $4.19 \pm 0.22$ | - |
| GT[2] | - | - | 1.74 | - | - | $3.84 \pm 0.19$ | $4.07 \pm 0.31$ | $4.16 \pm 0.19$ | - |
| VALL-E[1] | Encodec | train-28s | **4.05** | **0.740** | 0.442 | $3.86 \pm 0.19$ | $\mathbf{3.99 \pm 0.35}$ | $2.71 \pm 0.30$ | $2.66 \pm 0.31$ |
| VALL-E[2] | Encodec | train-54s | 6.03 | **0.741** | 0.414 | $3.86 \pm 0.20$ | $3.95 \pm 0.41$ | $\mathbf{2.61 \pm 0.25}$ | $\mathbf{2.60 \pm 0.28}$ |
| VALL-E[1] | Encodec | train-90s | 7.17 | 0.678 | 1.12 | $3.78 \pm 0.22$ | $3.74 \pm 0.48$ | $2.08 \pm 0.21$ | $2.11 \pm 0.22$ |
| VALL-E[2] | Encodec | train-180s | 95.46 | 0.711 | 12.77 | $3.83 \pm 0.21$ | $3.81 \pm 0.56$ | $1.74 \pm 0.19$ | $1.71 \pm 0.21$ |
| HALL-E[1] | MReQ-Encodec | train-90s | 4.63 | 0.701 | **0.196** | $\mathbf{3.88 \pm 0.19}$ | $3.84 \pm 0.33$ | $\mathbf{2.74 \pm 0.26}$ | $\mathbf{2.79 \pm 0.27}$ |
| HALL-E[2] | MReQ-Encodec | train-180s | **4.49** | 0.676 | **0.166** | $\mathbf{3.88 \pm 0.21}$ | $3.81 \pm 0.35$ | $2.58 \pm 0.27$ | $2.58 \pm 0.27$ |

Table 6: Results for Encodec and SpeechTokenizer. MinutesSpeech train-90s and test-90s are used for training and testing, respectively.

| TTS model | NAC model | WER↓ | SIM↑ | WD↓ | DNSMOS↑ | UTMOS↑ |
|---|---|---|---|---|---|---|
| GT | - | 10.30 | - | - | $3.79 \pm 0.23$ | $3.28 \pm 0.51$ |
| VALL-E | Encodec | 39.77 | **0.726** | 23.62 | $3.84 \pm 0.19$ | $3.61 \pm 0.48$ |
| HALL-E | MReQ-Encodec | **9.79** | 0.685 | 4.00 | $\mathbf{3.91 \pm 0.21}$ | $\mathbf{3.74 \pm 0.37}$ |
| VALL-E | SpeechTokenizer | 35.62 | **0.724** | 24.20 | $3.86 \pm 0.18$ | $3.65 \pm 0.38$ |
| HALL-E | MReQ-SpeechTokenizer | **9.12** | 0.678 | **3.09** | $\mathbf{3.95 \pm 0.19}$ | $\mathbf{3.77 \pm 0.33}$ |

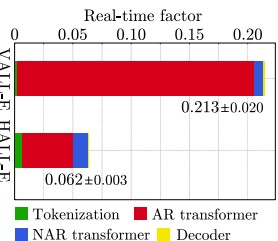

Figure 5: Real-time factor (RTF).

## 7.4 ANALYSIS AND ABLATION STUDIES

**Is the hierarchical structure essential?** The results in Table 7 demonstrate the impact of different frame rates adopted in MReQ on MinutesSpeech test-90s. For further details, please refer to Appendix C.1. From this table, it is evident that the proposed method achieves the best WER, indicating the importance of gradually increasing the frame rate in a hierarchical manner. We also observed a trade-off between WER and SIM, as the SIM deteriorates with a reduction in the number of 48Hz layers in the proposed method.

Table 7: Impact of varying hierarchical structure.

| Upsampling | WER↓ | SIM↑ |
|---|---|---|
| (8,48) | 10.27 | 0.693 |
| (8,16,48) | 10.71 | 0.693 |
| (8,16,24,48) | 9.79 | 0.685 |

**Long audio prompts.** Table 8 shows the SIM as a function of the audio length for prompt. The SIM values were calculated using audio segments longer than 65 seconds from the MinutesSpeech test-90s. As expected, longer audio length resulted in better performance. In zero-shot TTS, due to the limited context, the voice-cloning results are typically inferior to standard fine-tuning (Anastassiou et al., 2024). Our proposed method has the potential to push the limits of such zero-shot TTS capabilities.

Table 8: SIM as a function of audio prompt length.

| Len. | VALL-E | HALL-E |
|---|---|---|
| 3s | 0.727 | 0.685 |
| 10s | 0.801 | 0.750 |
| 20s | 0.830 | 0.809 |
| 40s | 0.851 | 0.846 |
| 60s | 0.856 | 0.859 |

**Ablation study for MReQ:** Table 9 shows the ablation study on the codec in the proposed method, using the LibriSpeech test set. The results indicate that the HSR loss most contributes to reconstruction performance. It is important to note that although the impact of the FLD loss may appear

minor, this does not mean that its influence on TTS is negligible. The codec model has a hierarchical structure with redundancy, meaning that information lost at the first layer can be compensated for in subsequent layers. However, when training the AR model, it is crucial for sufficient information to be present in the first layer, and the FLD loss is essential for achieving this. For further details, please refer to Appendix C.1.

Table 9: Ablation study for MReQ

| Model | WER↓ | PESQ↑ | UTMOS↑ |
|---|---|---|---|
| Proposed | 2.02 | $3.89_{\pm0.15}$ | $3.89_{\pm0.36}$ |
| w/o FLD loss | 2.03 | $3.89_{\pm0.15}$ | $3.87_{\pm0.36}$ |
| w/o HSR loss | 2.15 | $3.79_{\pm0.17}$ | $3.86_{\pm0.37}$ |
| w/o pre-training | 2.08 | $3.83_{\pm0.17}$ | $3.85_{\pm0.37}$ |

**Ablation study for HALL-E:** Table 10 shows the ablation study on HALL-E on MinutesSpeech test-90s. As shown, all the proposed methods were found to be effective. In particular, the results of training only with PreQ or PostQ highlight the importance of using the MRVQ sub-modules with NAR model as pointed out in Section 5. This underscores the necessity of

Table 10: Ablation study for HALL-E

| Model | WER↓ | SIM↑ | UTMOS↑ |
|---|---|---|---|
| Proposed | 9.79 | 0.685 | $3.74_{\pm0.37}$ |
| w/o pre-training | 49.80 | 0.647 | $2.55_{\pm0.30}$ |
| w/ PreQ only training | 10.36 | 0.682 | $3.52_{\pm0.38}$ |
| w/ PostQ only training | 10.11 | 0.683 | $3.73_{\pm0.36}$ |

designing the codec model with downstream tasks in mind, ensuring that it is optimized for the specific requirements of those tasks.

**Qualitative results.** Figure 6 shows the ground truth audio from MinutesSpeech test-90s and audio generated by each models. Both HALL-E and VALL-E models were trained on MinutesSpeech train-90s. As shown, the speech synthesized by HALL-E has duration similar to the ground truth, while VALL-E struggles with duration prediction. Training the model at a low frame rate not only enables the generation of long speech segments but also allows for capturing more natural long-term temporal dynamics. For more examples, please refer to Appendix D.1.

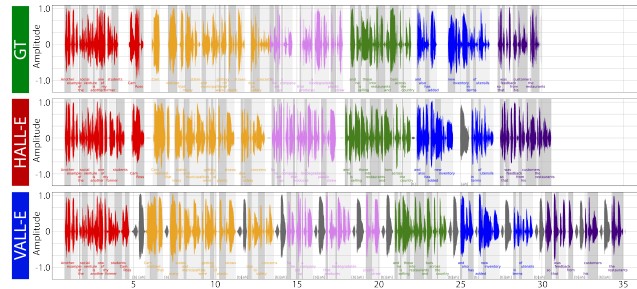

Figure 6: Samples of generated waveforms. Each sentence is displayed in a different color.

## 8 CONCLUSION

We introduced two novel approaches for minute-long zero-shot text-to-speech synthesis: MReQ and HALL-E. MReQ reduced the frame rate of the Encodec model to 8 Hz by reorganizing audio tokens via MRVQ. HALL-E efficiently synthesized minute-long speech by using 8Hz tokens in the AR model and MRVQ sub-modules in the NAR model. We demonstrated the effectiveness of these approaches on MinutesSpeech, the newly introduced dataset consisting of 40,000 hours of speech data. Our work contributed to promote zero-shot TTS research.

**Limitations and future work.** By reducing the frame rate to 8 Hz, our AR model can utilize longer contexts, enhancing the naturalness of the synthesized speech. We believe that to handle extended context is particularly advantageous for larger AR models such as AudioLM (Borsos et al., 2023), SpeechGPT (Zhang et al., 2023a), and PSLM (Mitsui et al., 2024). Demonstrating the effectiveness of our approach not only in TTS but also with these models remains a future work. Furthermore, as shown in Table 2, we have achieved shorter audio token length than the corresponding text token length. However, in our current AR model, we concatenate these tokens, which results in the text tokens becoming a bottleneck in terms of sequence length. Small-E (Lemerle et al., 2024) propose methods to mitigate this issue by processing each token individually and fusing them using cross-attention. Exploring such architectural enhancements is an important direction for future work. Lastly, as shown in Table 6, our method brought significant improvements with SpeechTokenizer, even more so than when applied to Encodec. It means that our approach can further enhance the objective of preserving linguistic information. This indicates that our method could serve as a replacement for traditional SSL models (Hsu et al., 2021; Chen et al., 2022) or ASR encoders (Lakhotia et al., 2021; Chu et al., 2023), marking an important direction for future research.

ACKNOWLEDGEMENT

This work was supported by JSPS KAKENHI Grant Numbers 22K12089 and 23H00490.

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

# A  MODEL DETAILS

## A.1  MReQ

Table 11: Hyper-parameters of Encodec

| Module | Hyper-Parameter | Values |
|---|---|---|
| Encoder & Decoder | ConvBlock Number | 4 |
| | ConvBlock Filter Sizes | [128, 256, 512, 1024] |
| | ConvBlock Strides | [10, 5, 5, 2] |
| | ConvBlock Kernel size | 7 |
| | ConvBlock Norm | Weight norm |
| | Activate Function | ELU (Clevert, 2015) |
| | LSTM number | 2 |
| RVQ | Codebook Number | 8 |
| | Codebook Dim | 128 |
| | Codebook Vocabulary | 1024 |
| Discriminator | MS-STFT (Défossez et al., 2022) | |
| | Window lengths | [2048, 1024, 512, 256, 128] |
| Loss Function | Adversarial Loss | 4 |
| Weight Coefficients | Feature Matching Loss | 4 |
| | L1 Loss | 0.1 |
| | MS-SPEC Loss (Yamamoto et al., 2020) | 2 |

Table 12: Hyper-parameters of MReQ-Encodec

| Module | Hyper-Parameter | Values |
|---|---|---|
| Sub-Encoders | ConvBlock Number | 2 |
| & Sub-Decoders | ConvBlock Filter Sizes | [512, 1024] |
| | ConvBlock Strides | |
| | $k = 1$ | [6, 1] |
| | $k = 2$ | [3, 1] |
| | $k = 3$ | [2, 1] |
| | ConvBlock Kernel size | 7 |
| | ConvBlock Norm | Weight norm |
| | Activate Function | ELU (Clevert, 2015) |
| | Bidiractional LSTM number | 2 |
| LRVQ | Codebook Number | $\alpha$-$\beta$-$\gamma$ |
| | $k = 1$ | 1-6-1 |
| | $k = 2$ | 2-6-2 |
| | $k = 3$ | 2-4-2 |
| | $k = 4$ | 3-0-0 |
| | Codebook Dim | 128 |
| | Codebook Vocabulary | 1024 |
| Loss Function | FLD Loss | $(\lambda^{\mathrm{FLD}}_{(1,1)}, \lambda^{\mathrm{FLD}}_{(2,3)}, \lambda^{\mathrm{FLD}}_{(3,5)}, \lambda^{\mathrm{FLD}}_{(4,8)}) = (8, 6, 4, 2)$ |
| Weight Coefficients | HSR Loss | $(\lambda^{\mathrm{HSR}}_1, \lambda^{\mathrm{HSR}}_2, \lambda^{\mathrm{HSR}}_3, \lambda^{\mathrm{HSR}}_4) = (8, 6, 4, 2)$ |

**Encodec:** Table 11 lists the various hyperparameters of the Encodec used in this study. Essentially, we utilized the configuration from the original Audiocraft (Copet et al., 2023b) implementation. The main difference is the modification of the strides within the encoder and decoder from the original [8, 5, 4, 2] to [10, 5, 5, 2]. This change increases the overall stride from 320 to 500, resulting in an improvement in the frame rate of the generated token sequences from 75Hz to 48Hz.

**MReQ-Encodec:** Table 12 presents the list of hyperparameters for the proposed MReQ-Encodec method. Modules added by MReQ, such as the sub-encoder, sub-decoder, and LRVQ, are the only

ones listed, as all other modules are identical to those in Encodec. Essentially, the parameters for the sub-encoder, sub-decoder, and LRVQ are based on the hyperparameters of the original encoder, decoder, and RVQ, respectively. We anticipate that applying MReQ to architectures other than Encodec will similarly yield stable hyperparameters by basing them on those of the original architecture, ensuring ease of adaptation.

In Section 4.2, we expressed the total loss function as $L_{\text{total}} = L_{\text{NAC}} + L_{\text{FLD}} + L_{\text{HSR}}$. The detailed formulation of $L_{\text{NAC}}$ in MReQ-Encodec is given by the following equation:

$$L_{\text{NAC}} = \lambda_t \cdot \ell_t(x_{\text{in}}, y) + \lambda_f \cdot \ell_f(x_{\text{in}}, y) + \lambda_g \cdot \ell_g(y) + \lambda_{feat} \cdot \ell(x_{\text{in}}, y) + \ell_w(w), \tag{18}$$

where each $\lambda$ represents the weight of its corresponding loss function. The terms $\ell_t$ and $\ell_f$ denote the reconstruction errors calculated using L1 loss and MS-SPEC loss, respectively. The terms $\ell_g$ and $\ell_{feat}$ correspond to the adversarial loss and the feature matching loss, respectively. The last term, $\ell_w$, represents the quantization errors across all quantizers, specifically PreQ, PostQ, and Quant.

**SpeechTokenizer:** In this paper, the implementation of SpeechTokenizer is almost identical to the Encodec architecture presented in Table 11. The only difference is that a bidirectional LSTM was used instead of an LSTM. For the teacher used in the teacher forcing process of Layer 1's RVQ, we followed the original paper and employed Hubert base ls960 [8].

In the original paper, 16,000Hz audio was used as the input data, with a stride of 320, resulting in a frame rate of 50Hz. Additionally, the SSL model typically employs a window size of 20ms with a 16,000Hz input, also yielding a frame rate of 50Hz. This setup enabled the application of the teacher forcing framework without any special adjustments in the temporal direction.

However, in this study, to ensure a fair comparison with Encodec, we adopted a 24,000Hz input and a stride of 500. Consequently, the frame rate became 48Hz, creating a mismatch with the 50Hz frame rate of the SSL model. To address this issue, we adjusted the audio input to the SSL model by speeding it up by a factor of $50/48$, ensuring that the duration of the SSL output aligns with that of the first layer of RVQ.

Moreover, while the original work utilized not only the MS-STFT discriminator employed by Encodec but also added a multi-period discriminator (MPD) and a multi-scale discriminator (MSD), as in HiFi-Codec (Yang et al., 2023a), it was observed that the discriminators were too strong under the aforementioned settings. As a result, using only the MS-STFT discriminator, similar to Encodec, yielded the best performance.

**MReQ-SpeechTokenizer:** As described above, the SpeechTokenizer ultimately uses an SSL model and, apart from the Bi-LSTM, matches the Encodec architecture in all respects. Therefore, when extending to MReQ, the architecture is identical to that of MReQ-Encodec, that is, the only difference between MReQ-SpeechTokenizer and MReQ-Encodec lies in the pretrained weights used.

In this study, we did not use SSL models to train the MReQ-SpeechTokenizer. This is because an SSL model was already utilized during the training of SpeechTokenizer, and since the features from that model were used for distillation, we deemed it unnecessary to reapply the SSL model.

A similar concept can be applied to models like Language-Codec (Ji et al., 2024). The goal of this model is to restrict the quantizers of the first three channels to learn only the compressed audio frame information in the specified space by applying masking during RVQ training. However, since the features obtained in this manner are used in MReQ training, there is no need to perform masking again. In other words, MReQ is flexible enough to be easily adapted and extended to new NAC models that may be developed in the future to learn better features.

### A.2 HALL-E

**VALL-E:** Table 13 presents the hyperparameters of the AR LM in the VALL-E model. For this implementation, we based our work on the LM of Audiocraft's MusicGen (Copet et al., 2023b) and referred to an unofficial VALL-E implementation [9].

Table 14 presents the hyperparameters of the NAR LM in the VALL-E model. The fundamental hyperparameters are similar to those of the AR LM, with differences in the values for Dim, Heads,

---

[8] https://huggingface.co/facebook/hubert-base-ls960
[9] https://github.com/lifeiteng/vall-e

Table 13: Hyper-parameters of AR LM in VALL-E

| Module | Hyper-Parameter | Values |
|---|---|---|
| Transformer | Dim | 1280 |
| | Number of Heads | 20 |
| | Number of Layers | 36 |
| | Dim of FFN | 5120 |
| | Norm | Layer norm |
| | Activate Function | GELU (Hendrycks & Gimpel, 2016) |
| | Positional Encoding | sin (Vaswani et al., 2017) |
| | Text Tokenizer | Byte-Pair Encoding (BPE) (Sennrich, 2015) |
| | Text Vocab | 258 |

Table 14: Hyper-parameters of NAR LM in VALL-E

| Module | Hyper-Parameter | Values |
|---|---|---|
| Transformer | Dim | 1024 |
| | Number of Heads | 16 |
| | Number of Layers | 24 |
| | Dim of FFN | 4096 |
| | Norm | Layer norm |
| | Activate Function | GELU (Hendrycks & Gimpel, 2016) |
| | Positional Encoding | sin (Vaswani et al., 2017) |
| | Text Tokenizer | Byte-Pair Encoding (BPE) (Sennrich, 2015) |
| | Text Vocab | 258 |

and Layers, resulting in a smaller number of parameters compared to the AR LM. It is generally known that the task of generating acoustic tokens from acoustic input tokens handled by the NAR LM is simpler than the task of generating tokens, including linguistic elements, from text handled by the AR LM. Indeed, in our preliminary experiments, increasing the model size of the NAR LM to match that of the AR LM resulted in minimal differences in performance.

**HALL-E:** The architecture of the AR LM in HALL-E is almost identical to that of VALL-E, as shown in Table 13. The only difference lies in the input tokens being fed in a delayed format. This method was introduced in MusicGen and adopted in TTS systems such as Lyth & King (2024). By employing this approach, it becomes possible to efficiently train multi-layer token sequences within the framework of next-token prediction.

In HALL-E's AR LM, although it operates at a very low frame rate of 8Hz, it needs to handle six layers of tokens. If this were trained in a flattened format, the sequence length would be equivalent to 48Hz, rendering the frame rate reduction meaningless. However, by handling it in a delayed format, the sequence length can be maintained at nearly the 8Hz frame rate, allowing for the optimal utilization of tokens obtained through MReQ.

The architecture of HALL-E's NAR model similarly shares many points with VALL-E's NAR LM, as shown in Table 14. There are two major differences: 1) a cross-attention layer is additionally inserted into each transformer layer, using text tokens as input for the cross-attention, and 2) as described in Section 5, the output tokens of PostQ and PreQ are combined for input and output.

**1)** In this study, HALL-E handles very long audio segments, such as 90s or 180s. However, as the NAR LM processes tokens at 48Hz, it is difficult to train with all the tokens as input. Therefore, we consider training with only a portion of these long tokens. Specifically, by clipping the audio length corresponding to 90s or 180s so that it becomes 28s or 54s, it is possible to train sequences of the same length as VALL-E's NAR LM. However, VALL-E's NAR LM requires the text corresponding to the audio tokens to be concatenated with the audio tokens as input. In this study, where there is no alignment between the text and audio, it is difficult to extract the corresponding text portion for the clipped audio tokens and concatenate them.

To address this, we stopped using the traditional concatenation method for inputting text information and instead used cross-attention to input the text information. This allows the attention mechanism

to automatically learn the alignment with the input audio tokens. To our knowledge, no model in the VALL-E framework has incorporated this kind of cross-attention mechanism to add text information. Small-E (Lemerle et al., 2024) introduces this idea in a more refined form with Linear Causal Language Models.

During inference, only audio up to 28s or 54s can be processed at most, so a method is employed where new audio is generated sequentially by sliding 5s at a time. For more detailed inference algorithms and other information, please refer to our publicly available GitHub repository[10].

**2)** The technique of appropriately combining PostQ and PreQ for input, introduced in Section 5, was implemented with the aim of promoting learning by bringing the structure closer to the original form of the VALL-E NAR LM, as explained in that section. While this is indeed another major difference from VALL-E, it should be noted that this operation involves using the modules acquired through MReQ to properly transform tokens outside of the NAR LM, and no changes have been made to the architecture of the NAR LM itself.

## B  DATASET DETAILS

### B.1  MINUTESSPEECH TEST SET DETAILS

As mentioned in Section 6, the test set was gathered from broadcasts available under a Creative Commons license. The goal was to collect redistributable data to facilitate future research on long-form speech generation. The collection method involved first filtering English podcast descriptions for those containing Creative Commons-related terms. Subsequently, podcasts explicitly labeled with Creative Commons were manually selected. In most cases, the Creative Commons license applied only to the music used in the broadcasts, with few instances of the license covering the broadcasts themselves. Next, ASR was used to generate provisional transcripts, which were then refined by native English speakers. Simultaneously, the quality of the audio was evaluated, and broadcasts with high audio quality, suitable for speech synthesis, were selected. The list of podcast broadcasts included in the final dataset is presented in Table 15.

Table 15: The list of podcast broadcasts under the Creative Commons license included in the MinutesSpeech test set

| URLs | Total Duration (h) | License |
|---|---|---|
| `https://anchor.fm/s/6f51ed88/podcast/rss/` | 6.0 | CC BY 4.0 |
| `https://feeds.captivate.fm/research-culture//` | 1.8 | CC BY-SA 4.0 & CC BY-ND 4.0 |
| `https://anchor.fm/s/640d7168/podcast/rss/` | 0.9 | CC BY-NC-ND 4.0 |
| `https://feeds.hubhopper.com/b38dfaea594789a83b87cb96d3f79004.rss` | 0.4 | CC BY-NC |

### B.2  EXAMPLES OF THE TRANSCRIPTS

Table 16 shows a portion of an actual transcript. This example from the MinutesSpeech test-90s was generated by combining segments using ASR timestamps and speaker identification annotations by native English speakers, ensuring that the maximum duration does not exceed 90 seconds. As illustrated in this example, podcasts often feature a dialogue format between two individuals, which leads to variability in the duration of a single speaker's continuous utterance. In fact, even within this table, the durations range from a minimum of 3.68 seconds to a maximum of 79.82 seconds. For the complete dataset and further details, please refer to `https://github.com/YutoNishimura-v2/HALL-E`.

---

[10]`https://github.com/YutoNishimura-v2/HALL-E`

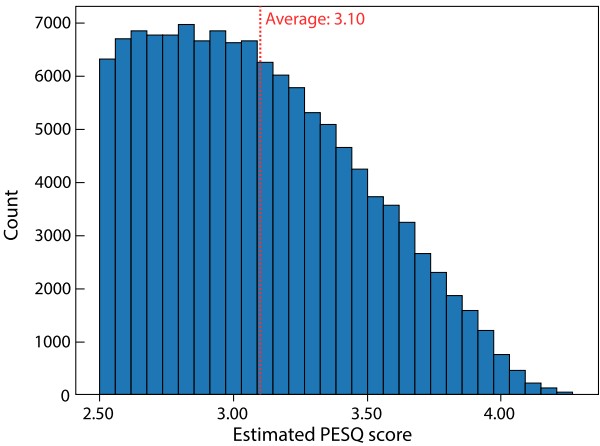

Figure 7: Distribution of estimated PESQ scores.

### B.3 QUALITY

Figure 7 shows the distribution of PESQ scores estimated by TorchAudio-Squim Kumar et al. (2023a) for the MinutesSpeech training dataset, with an average score of 3.10. Compared to LibriSpeech, which has an average score of 3.56, MinutesSpeech has lower-quality audio data. However, audio reconstructed by the Encodec and SpeechTokenizer models trained on MinutesSpeech achieved scores of 3.45 and 3.52, respectively, as shown in Table 3. This indicates that the dataset possesses sufficient quality for TTS purposes.

## C ABLATION STUDIES

### C.1 MREQ

**Further information of the ablation study for a hierarchical structure in MReQ.** In Section 7.4, we demonstrated that the proposed method achieves better results compared to other upsampling approaches. The parameters used are as follows: For (8, 48), $K = 2$, $(\alpha_1, \beta_1, \gamma_1) = (1, 6, 1)$, $(\alpha_2, \beta_2, \gamma_2) = (7, 0, 0)$. For (8, 16, 48), $K = 3$, $(\alpha_1, \beta_1, \gamma_1) = (1, 6, 1)$, $(\alpha_2, \beta_2, \gamma_2) = (2, 6, 2)$, $(\alpha_3, \beta_3, \gamma_3) = (5, 0, 0)$.

Additionally, there is an important point that needs to be addressed. when the upsampling rate is (8, 48), the loss diverged after about 60% of training. This is likely due to the instability caused by the difference in the 1st layer between the Encodec used for the pre-training and MReQ-Encodec used for the post-training, which led to unstable learning. This highlights the importance of hierarchical modeling to prevent sudden distribution shifts.

**Is FLD loss needed?** Table 17 compares the case where MReQ-Encodec was trained without using the FLD loss and HALL-E was trained using it. Here, MinutesSpeech-90s was used for evaluation. Interestingly,

Table 17: Ablation study for FLD loss in MReQ

| Model | WER↓ | SIM↑ | WD↓ | DNSMOS↑ |
|---|---|---|---|---|
| HALL-E | 9.79 | 0.685 | **4.00** | **3.91** ± 0.21 |
| w/o FLD loss | **9.07** | **0.704** | 4.51 | 3.88 ± 0.20 |

when the FLD loss is excluded, the results do not worsen across all metrics—WER and SIM actually improve. However, WD and DNSMOS deteriorate to a non-negligible extent. This indicates that using the FLD loss allows for generating more natural, human-like speech. Below, we will discuss the reasons for this.

According to preliminary experiments, when MReQ is trained entirely without a pre-trained model, LRVQ at frame rates like 8Hz is ignored, and the learning process progresses to supplement all the information at higher frame rates. Since post-training with pre-trained model weights is applied here, this issue is relatively mitigated, and relatively good results are achieved even without the FLD loss.

On the other hand, when the frame rate is compressed, there is a tendency for finer temporal information to be lost. This means, in principle, that acoustic information is more likely to be lost than

Table 16: Example transcript from `https://anchor.fm/s/6f51ed88/podcast/rss/`

| Start (s) | End (s) | Duration (s) | Transcript |
|---|---|---|---|
| 8.11 | 27.47 | 19.36 | Welcome to Deconstructing Management, a podcast made by college students for college students. We've interviewed the chapter authors of the OpenStack's Principles of Management textbook... |
| 34.57 | 92.55 | 57.98 | I'm here with Chapter 7 author, Siri Terjesen. Dr. Terjesen holds a postdoctoral degree from the Queensland University of Technology... |
| 96.19 | 99.99 | 3.80 | Very good, Eric. Thanks so much for inviting me to join you in your class today. |
| 100.53 | 104.65 | 4.12 | Do you believe that successful entrepreneurs are born, or are they made? |
| 104.65 | 110.61 | 5.96 | That's a great question. And I think there are quite a lot of entrepreneurs that are born, right? |
| 123.79 | 142.33 | 18.54 | And we didn't instigate that. But I think that they can also be made... |
| 156.81 | 168.57 | 11.76 | I would have to agree with you because they could be born or, like my cousin... |
| 172.33 | 180.27 | 7.94 | He's going to start, like a party business, kind of like where he has bouncy houses... |
| 180.85 | 252.06 | 71.21 | That's neat. So really your cousin is a great example of a serial or even a portfolio entrepreneur... |
| 252.06 | 259.86 | 7.80 | So going straight to another question, what are the most important traits a person must have to become a successful entrepreneur? |
| 260.12 | 313.72 | 53.60 | That's a great question, Eric. And you may have seen this in your cousin, right?... |
| 318.13 | 357.97 | 39.84 | Well, you're never too old! So there's a huge opportunity there... |
| 365.94 | 445.76 | 79.82 | That's a great question. So, I think folks who have low-risk preferences may not make good entrepreneurs... |
| 459.27 | 526.33 | 67.06 | That's the first step, right? It's thinking about what the business would be... |
| 527.01 | 541.68 | 14.67 | It's mainly, it's also like, to start with like, shirts, like something so simple, like shirts... |
| 544.02 | 547.70 | 3.68 | So, so like, the controllers for like, Xbox, and like, PS4. |

linguistic information. The first layer of Encodec's RVQ, which serves as the pre-trained model, contains not only linguistic information but also a rich amount of acoustic information. Therefore, applying the FLD loss facilitates learning that retains both linguistic and acoustic information. This is believed to contribute to more natural speech generation. In contrast, when the FLD loss is not used, more acoustic information is lost. As a result, while the retention of linguistic information may improve WER, the overall outcome is less natural.

For this reason, we have adopted the FLD loss, which enables the generation of more natural speech. Further improvements in WER and SIM while using this method remain a task for future work.

**Is it possible to use an even lower frame rate?** Table 18 shows the results of HALL-E trained using a lower frame rate adopted in MReQ. The evaluation was conducted similarly using MinutesSpeech-90s. Here, a very low frame rate of 4Hz was applied to the first LRVQ quantization. As the results indicate, this caused a significant deterioration in WER. In general, the average duration of each

phoneme is known to be around 100ms, which corresponds to approximately 10Hz. Therefore, using a frame rate as low as 4Hz is likely to be fundamentally challenging.

On the other hand, we can see a substantial improvement in WD. This suggests that lowering the frame rate to 4Hz makes it easier to learn longer-term temporal dynamics, enabling the generation of more natural durations. From this, it is evident that while further lowering the frame rate involves significant difficulties, the potential benefits are also considerable.

Table 18: Ablation study for lower frame rate

| Upsampling | WER↓ | SIM↑ | WD↓ | DNSMOS↑ |
|---|---|---|---|---|
| $(8, 16, 24, 48)$ | **9.79** | 0.685 | 4.00 | **3.91** $_{\pm 0.21}$ |
| $(4, 8, 16, 24, 48)$ | 20.07 | **0.690** | 1.95 | 3.90 $_{\pm 0.21}$ |

**Is it possible to reduce a bps?** In MReQ, the number of Quant layers can be freely adjusted. Therefore, we considered reducing the number of Quant layers by half. In the original MReQ, the bitrate is set to match that of Encodec, which is 3.84 kbps, but by reducing the number of Quant layers in this way, it can be decreased to 2.64 kbps. Table 19 shows the performance changes of MReQ on the LibriSpeech test set when the bitrate is reduced in this manner. As the table indicates, significant performance degradation occurs even for a relatively simple task like Speech Reconstruction. This suggests that it is preferable to reduce the bitrate through improvements on the NAC model side, rather than by reducing the frame rate within the MReQ framework.

Table 19: Hyperparameter study for lower bps in MReQ

| $\beta$s | WER↓ | PESQ↑ |
|---|---|---|
| $(6, 6, 4, 0)$ | **2.02** | **3.47** |
| $(3, 3, 2, 0)$ | 2.13 | 3.42 |

Table 20 shows the performance of HALL-E using this MReQ evaluated on the MinutesSpeech-90s dataset. As indicated by the WER degradation already observed in MReQ, WER worsens significantly when training HALL-E. On the other hand, improvements are seen in all other metrics besides WER. This can be attributed to the fact that the reduction in the number of layers handled by the AR LM—from 6 layers to 3—makes the training process much easier, allowing for better learning of aspects like duration. Additionally, reducing the number of Quant layers, and thereby the amount of information contained in each block, results in smaller relative differences between blocks, making it easier for the NAR LM to learn, which likely contributes to improvements in metrics like SIM. However, the deterioration in WER is too significant to ignore, indicating that it is generally better to maintain the same bitrate.

Table 20: Ablation study for lower bps in HALL-E

| $\beta$s | WER↓ | SIM↑ | WD↓ | DNSMOS↑ |
|---|---|---|---|---|
| $(6, 6, 4, 0)$ | **9.79** | 0.685 | 4.00 | 3.91 $_{\pm 0.21}$ |
| $(3, 3, 2, 0)$ | 13.31 | **0.713** | 2.76 | **3.92** $_{\pm 0.21}$ |

## C.2 HALL-E

Table 21: Zero-shot TTS performances of HALL-E with different $\alpha$-$\beta$-$\gamma$ pairs

| Dataset | TTS model | $(\alpha_1, \alpha_2, \alpha_3, \alpha_4)$ | WER↓ | SIM↑ | WD↓ | DNSMOS↑ |
|---|---|---|---|---|---|---|
| | GT | - | 10.30 | - | - | 3.79 $_{\pm 0.24}$ |
| | HALL-E | $(1, 1, 2, 4)$ | 9.77 | 0.688 | 4.25 | 3.92 $_{\pm 0.23}$ |
| MinutesSpeech-90s | HALL-E | $(1, 1, 3, 3)$ | **9.60** | **0.702** | 4.59 | **3.93** $_{\pm 0.22}$ |
| | HALL-E | $(1, 2, 2, 3)$ | 9.79 | 0.685 | **4.00** | 3.91 $_{\pm 0.21}$ |
| | HALL-E | $(1, 2, 3, 2)$ | 10.58 | 0.696 | 4.09 | 3.90 $_{\pm 0.19}$ |
| | GT | - | 1.74 | - | - | 3.84 $_{\pm 0.19}$ |
| | HALL-E | $(1, 1, 2, 4)$ | **4.37** | **0.706** | 0.201 | **3.89** $_{\pm 0.19}$ |
| LibriSpeech | HALL-E | $(1, 1, 3, 3)$ | 4.67 | 0.703 | **0.165** | 3.88 $_{\pm 0.20}$ |
| | HALL-E | $(1, 2, 2, 3)$ | 4.63 | 0.701 | 0.196 | 3.88 $_{\pm 0.19}$ |
| | HALL-E | $(1, 2, 3, 2)$ | 5.68 | 0.688 | 0.178 | **3.89** $_{\pm 0.19}$ |

**Is different $\alpha$-$\beta$-$\gamma$ pair more effective?** Table 21 shows the performance changes when the $\alpha$-$\beta$-$\gamma$ combinations used in each LRVQ of MReQ are modified. It should be noted that in all cases, the bps is adjusted to be equal to that of Encodec, and $\beta_4 = \gamma_4 = 0$. As the results indicate, there is no consistently strong combination across all metrics. On the other hand, when $\alpha_4 = 2$, the results are consistently the worst. This implies that reducing the number of layers corresponding to 48Hz too

much leads to performance degradation. As shown in Figure 1, significant performance deterioration starts from 48Hz in Encodec, suggesting that this layer should be retained as much as possible.

In this study, we propose $(1, 2, 2, 3)$ as the recommended method, as it consistently achieved balanced and good results across both datasets. However, other configurations also produced satisfactory results, demonstrating the robustness of our method to variations in hyperparameters.

## D  MORE QUALITATIVE RESULTS

### D.1  DURATION ANALYSIS

Figure 8 shows an qualitative example with manually annotated segments for each word. In the first sentence, highlighted in red, the duration of the proper noun "Cam Ross" differs significantly across the models. In the speech generated by HALL-E, this phrase is pronounced more slowly, closely resembling the ground truth. This effect can be attributed to the low sampling rate tokens, which likely made it easier to learn contextual relationships between words. Additionally, VALL-E tends to introduce more filler sounds and breaths in the latter part of the utterance, potentially disrupting the duration prediction.

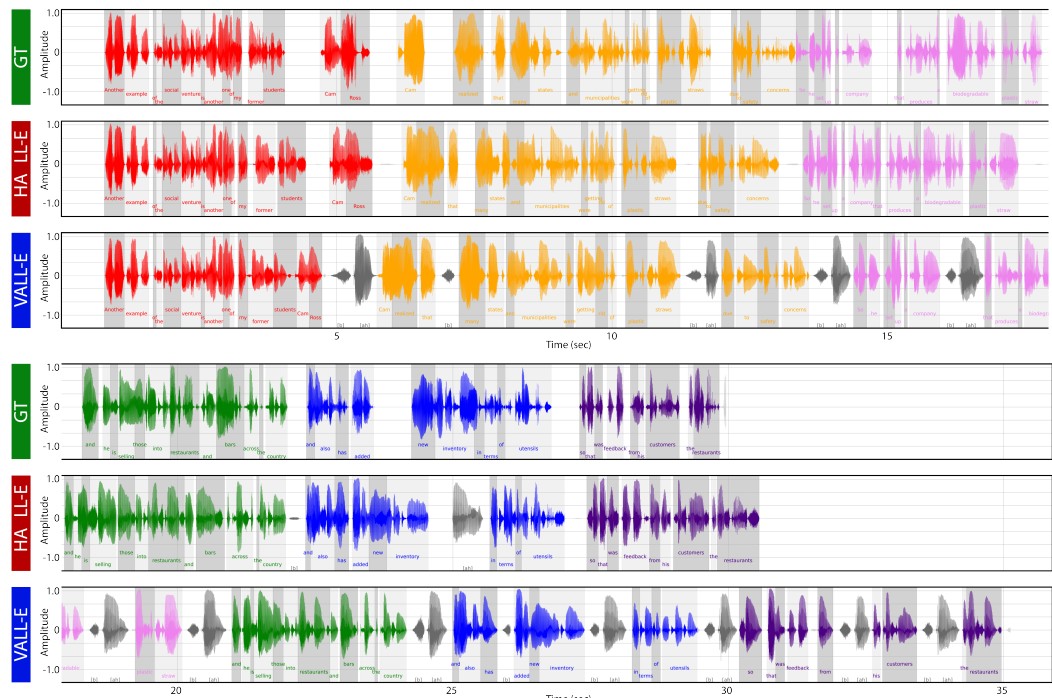

Figure 8: Qualitative example with transcription (zoom in to view details).

Figure 9 shows the waveform of a different sample compared to Figure 6. Similar to the observations in Figure 6, VALL-E generates an unnatural waveform with almost no silence, while HALL-E produces a more natural waveform, closely resembling the ground truth with appropriate amounts of silence.

Figure 10 presents the duration distribution for the entire dataset, calculated from MinutesSpeech test-90s and test-180s. As consistently demonstrated by the WD in Table 4, HALL-E's distribution is consistently closer to the GT compared to VALL-E. In addition to the issue of failing to insert silent pauses, VALL-E tends to overly repeat breathing sounds and fillers, resulting in longer durations. HALL-E, by utilizing a very small frame rate of 8Hz, overcomes these issues specific to AR models and achieves a more natural duration.

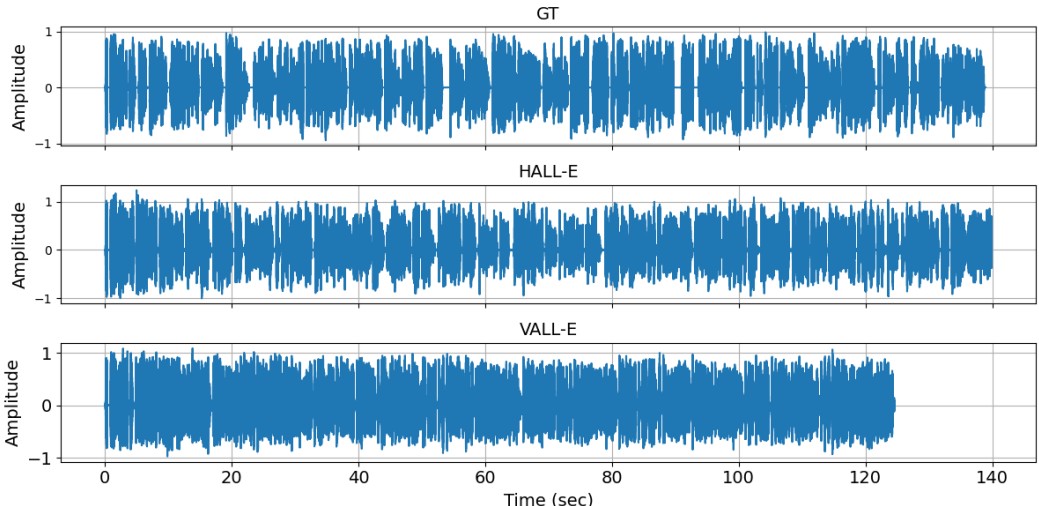

Figure 9: A sample from MinutesSpeech test-180s. Both the HALL-E and VALL-E models were trained using MinutesSpeech train-180s.

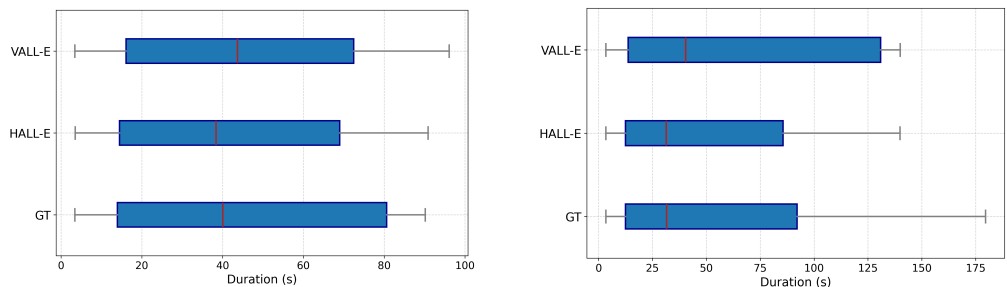

Figure 10: Comparison of the distribution of generated speech durations across different methods in the MinutesSpeech test sets. The left plot shows the distribution for the test-90s set, while the right plot shows the distribution for the test-180s set. Both the HALL-E and VALL-E models were trained using the corresponding training sets (train-90s and train-180s).

Table 22: The comparison of filler generation abilities between HALL-E and VALL-E. The left table presents the analysis results for the MinutesSpeech test-90s, while the right table shows the results for the test-180s. In each case, HALL-E and VALL-E are trained on the corresponding train-90s and train-180s datasets.

| TTS model | Avg. Fillers per text | Total Fillers |
|---|---|---|
| GT | $3.74 \pm 3.66$ | 2567 |
| HALL-E | $\mathbf{3.45} \pm \mathbf{3.35}$ | **2364** |
| VALL-E | $3.22 \pm 3.30$ | 2207 |

| TTS model | Avg. Fillers per text | Total Fillers |
|---|---|---|
| GT | $4.64 \pm 5.22$ | 2493 |
| HALL-E | $\mathbf{4.19} \pm \mathbf{4.66}$ | **2254** |
| VALL-E | $3.87 \pm 7.05$ | 2081 |

## D.2 FILLER ANALYSIS

Fillers are an essential component for generating more natural human-like speech (Mitsui et al., 2023). In this study, we propose and use the MinutesSpeech dataset collected from podcasts. This dataset includes numerous instances of human dialogue, naturally containing many fillers. The ability to generate fillers at a frequency similar to human speech could be considered a future benchmark for TTS systems.

Here, using the MinutesSpeech test set, we demonstrate the spontaneous speech generation capability of HALL-E by analyzing and comparing the frequency of filler words in both ground truth and generated speech. The list of filler words present in the MinutesSpeech test set is shown in Table 23.

Table 23: Filler Words List used in MinutesSpeech test set

| The list of filler words | | | | | | | | | | | |
|---|---|---|---|---|---|---|---|---|---|---|---|
| so | you know | like | actually | right | well | i mean | um | you see | basically | literally | uh |

Table 22 shows the statistics of fillers in each MinutesSpeech test set. In all cases, HALL-E generates fillers at a frequency closer to the ground truth compared to VALL-E. This improvement is likely due to the use of an 8Hz frame rate, which facilitates better language understanding in the AR LM.

Notably, in the 180s results for VALL-E, the standard deviation is significantly large. This is because VALL-E, as an AR model, sometimes exhibits repeated generation, producing fillers consecutively. Such cases were not observed in HALL-E, suggesting that a lower frame rate is crucial for suppressing repeats.

However, there is still a gap between the generated results and the GT. This is likely because more advanced role-playing capabilities are needed. It is expected that utilizing models such as SLMs, which extend LLMs to speech, could enable more human-like filler generation with their advanced language processing abilities.

## E  COMPARISON WITH NAR TTS

Table 24 analyzes the performance of MaskGCT (Wang et al., 2024b) by speech duration using the pre-trained model provided by the authors. We observed that the performance of MaskGCT significantly decreases as the length of synthesized speech increases. This indicates that even with NAR TTS, generating long speech in a single inference procedure is challenging. Tables 25 and 26 compare VALL-E, HALL-E and MaskGCT in the test-30s and test-90s settings, respectively. In the 30-second test setting, the WER of MaskGCT is lower than that of VALL-E and HALL-E. However, its UTMOS is worse compared to HALL-E. In the 90-second test setting, only HALL-E achieves a WER lower than 10.0%, along with the highest UTMOS. These results highlight the effectiveness of the proposed method.

Table 24: MaskGCT performance by test duration.

| TTS Model | Duration (sec) | WER↓ | SIM↑ | DNSMOS↑ | UTMOS↑ |
|---|---|---|---|---|---|
| MaskGCT | 30 | 7.74 | 0.763 | $3.95 \pm 0.23$ | $3.42 \pm 0.46$ |
| MaskGCT | 45 | 10.75 | 0.763 | $3.94 \pm 0.22$ | $3.39 \pm 0.49$ |
| MaskGCT | 60 | 23.08 | 0.757 | $3.92 \pm 0.23$ | $3.30 \pm 0.54$ |
| MaskGCT | 90 | 51.12 | 0.691 | $3.57 \pm 0.51$ | $2.57 \pm 1.02$ |

Table 25: Performance comparison on test-30s.

| TTS Model | WER↓ | SIM↑ | DNSMOS↑ | UTMOS↑ |
|---|---|---|---|---|
| GT | 10.65 | - | $3.74 \pm 0.29$ | $3.20 \pm 0.51$ |
| VALL-E (train-28s) | 11.84 | 0.731 | $3.84 \pm 0.19$ | $3.61 \pm 0.41$ |
| VALL-E (train-90s) | 14.83 | 0.695 | $3.82 \pm 0.20$ | $3.46 \pm 0.50$ |
| HALL-E (train-90s) | 10.88 | 0.685 | $3.87 \pm 0.21$ | $3.61 \pm 0.38$ |
| MaskGCT | 7.74 | 0.763 | $3.95 \pm 0.23$ | $3.42 \pm 0.46$ |

Table 26: Performance comparison on test-90s.

| TTS Model | WER↓ | SIM↑ | DNSMOS↑ | UTMOS↑ |
|---|---|---|---|---|
| GT | 10.30 | - | $3.79 \pm 0.24$ | $3.28 \pm 0.51$ |
| VALL-E (train-28s) | 39.77 | 0.726 | $3.84 \pm 0.19$ | $3.61 \pm 0.48$ |
| VALL-E (train-90s) | 16.14 | 0.712 | $3.87 \pm 0.17$ | $3.58 \pm 0.60$ |
| HALL-E (train-90s) | 9.79 | 0.685 | $3.91 \pm 0.21$ | $3.74 \pm 0.37$ |
| MaskGCT | 51.12 | 0.691 | $3.57 \pm 0.51$ | $2.57 \pm 1.02$ |

