# OpenReview forum: "HALL-E: Hierarchical Neural Codec Language Model for Minute-Long Zero-Shot Text-to-Speech Synthesis"
_ICLR.cc/2025/Conference — ICLR 2025 Poster_

### Official Review · Reviewer_JBx3 · 2024-10-28

**Soundness:** 3
**Presentation:** 3
**Contribution:** 3
**Rating:** 6
**Confidence:** 5

**Summary:**

The paper introduces a novel text-to-speech (TTS) model designed to generate extended speech sequences. To address the inherent limitations of autoregressive language models in handling long outputs, the authors propose the use of *multi-resolution audio codes* (MReQ). Furthermore, the paper presents HALL-E, an autoregressive language model specifically developed to generate these MReQ audio codes efficiently. In addition to the model, the authors construct a new benchmark dataset comprising 40,000 hours of curated speech data. The experimental results demonstrate that the proposed approach can achieve a low frame rate of 8 Hz, highlighting its potential for generating high-quality long-form speech with reduced computational overhead.

**Strengths:**

1. This paper introduces the Multi-Resolution Requantization (MReQ) model, designed to enhance signal processing across multiple granularities.
2. It presents the development of HALL-E, an autoregressive language model capable of generating extended speech waveforms with high fidelity.
3. Furthermore, the study proposes a novel benchmark dataset, MinutesSpeech, specifically curated to evaluate the performance of models on long-duration speech sequences.

**Weaknesses:**

1. The concept of time-varying audio codes is not a new development. For reference, please see the work by [Variable-rate Discrete Representation Learning (2021)] [https://arxiv.org/pdf/2103.06089].

2. A notable issue lies in maintaining temporal coherence and acoustic fidelity, as evidenced by the declining performance observed in the SIM score.

3. The model continues to face difficulties in handling spontaneous speech, highlighting an ongoing challenge in this area.

**Questions:**

--

---

> ### Author Response · Authors · 2024-11-23
>
> Thank you very much for your positive feedback and constructive comments.
> Below, we respond to each comment.
>
> **Comment 1:** The concept of time-varying audio codes is not a new development. For reference, please see the work by [Variable-rate Discrete Representation Learning (2021)] [https://arxiv.org/pdf/2103.06089].
>
> **Response:** We have added the suggested paper to the reference list and updated Sections 2 accordingly.
>
> **Comment 2:** A notable issue lies in maintaining temporal coherence and acoustic fidelity, as evidenced by the declining performance observed in the SIM score.
>
> **Response:** We conducted an additional experiment to demonstrate that providing longer audio prompts can mitigate the issue and significantly improve the SIM score. As shown in Table 8, with a prompt length of 60 seconds, HALL-E slightly outperformed VALL-E. Since the reduced frame rate achieved by MReQ enables efficient generation, even with long prompts of 60 seconds, this result demonstrates that HALL-E is capable of achieving high-fidelity synthesis, highlighting its practical value.
>
>
> **Table 8. SIM as a function of audio prompt length.**
>
> | Prompt length        | VALL-E            | HALL-E |
> |---------------|---------------|---------------|
> | 3s (default)            | 0.727 | 0.685 |
> | 10s                     | 0.801 | 0.750 |
> | 20s                     | 0.830 | 0.809 |
> | 40s                     | 0.851 | 0.846 |
> | 60s                     | 0.856 | 0.859 |
>
>
> **Comment 3:** The model continues to face difficulties in handling spontaneous speech, highlighting an ongoing challenge in this area.
>
> **Response:** We agree with the reviewer that handling spontaneous speech remains a significant challenge. Improving prosody modeling within the AR models is essential for achieving greater naturalness in synthesized spontaneous speech. We believe that low frame rate tokens will be key to addressing this challenge, and that the MinuteSpeech dataset and HALL-E serve as valuable resources to facilitate research in this direction.

---

> > ### Comment · Reviewer_JBx3 · 2024-11-25
> >
> > Thank you for addressing my concerns and providing detailed clarifications. I particularly appreciate the additional analyses you conducted on SIM for larger audio prompts, which I found to be highly insightful and valuable.

---

### Official Review · Reviewer_aZfo · 2024-10-29

**Soundness:** 3
**Presentation:** 3
**Contribution:** 4
**Rating:** 8
**Confidence:** 4

**Summary:**

The paper presents three contributions: (1) Multi-Resolution Requantization (MReQ), a framework that post-trains a conventional RVQ codec into one with a different token rate in each layer; (2) HALL-E, an extension of the VALL-E framework to utilize an MReQ-based codec; and (3) the introduction of MinutesSpeech, a new benchmark for minute-long speech synthesis. Leveraging the MReQ concept, the authors successfully train a codec whose first layer operates at only 8Hz. With this low-token rate modeling, the HALL-E model generates minutes-long speech with significantly better WER and RTF compared to the original VALL-E model.

**Strengths:**

The three contributions listed above are valid and well-received. The topic of minute-long speech synthesis is also one that will attract a wide audience.

**Weaknesses:**

While the paper is generally well-written, there are several points that seem unclear to me. It is difficult to fully accept their proposal unless these points are clarified. I am open to reconsidering the score later.

- Unreasonable usage of PESQ raises doubts about their experiments.
  - Why and how can PESQ be used to filter out noisy data in the data curation pipeline? PESQ requires reference audio to compute the score, and I cannot understand how it could be applied for data curation purposes.
  - Why is the PESQ score for the ground truth so low in Table 3? The PESQ score for the ground truth should be 4.64.
- Questionable inference and evaluation settings
  - Zero-shot TTS experiment was conducted by “continual” setting or “cross-utterance” setting? From the description, it’s unclear.
  - Assuming that a "continual" setting is used, did the authors exclude the audio prompt portion of the synthesized audio when computing the speaker SIM? This is the approach taken in the VALL-E paper and other related works. The increase in speaker SIM in Table 9 with respect to the prompt length is unusually large, raising concerns that the authors may not have excluded the prompt from the speaker similarity computation.
  - Why is DNSMOS used? DNSMOS is specifically designed and trained to assess the cleanness of speech rather than its naturalness. I strongly suggest replacing it with a naturalness MOS metric, such as UTMOS.
- Other minor request to assess the quality of the model
  - Please present the speaker similarity and MOS scores (UTMOS?) for Table 3 and Table 10.
  - Please provide a breakdown of the AR and NAR model RTFs for Table 7.

Additionally, I found several minor issues in the description. They are not critical, but they need to be revised:
  - Equation 2 is likely incorrect. I believe it should be x_l = x_{l-1} - \tilde{z_l}
  - Similarly, Equation 9 is likely incorrect.
  - In Table 8, the upsampling ratio should be explicitly described instead of simply listing "HALL-E" in the last row.

**Questions:**

Please address the questions raised in the weaknesses section.

---

> ### Author Response · Authors · 2024-11-23
> **Reply to Reviewer aZfo (Part 1)**
>
> Thank you very much for your insightful review and positive feedback. Below, we respond to each question and comment.
>
> **Question 1:** Why and how can PESQ be used to filter out noisy data in the data curation pipeline?
>
> **Response:** Following the previous study [1], PESQ scores are estimated using TorchAudio-Squim [2], which provides a reference-less method for predicting speech quality metrics. This is the reason why PESQ can be used in the data curation pipeline. We have clarified this in the revised paper.
>
> The score distribution for the MinutesSpeech training dataset is shown in Figure 7 (Appendix B.3), with an average score of 3.10.
> Compared to LibriSpeech, which has an average score of 3.56, MinutesSpeech has lower-quality audio data.
> However, audio reconstructed by the Encodec and SpeechTokenizer models trained on MinutesSpeech achieved scores of 3.45 and 3.52, respectively. This indicates that the dataset possesses sufficient quality for TTS purposes.
>
> [1] A. Vyas, et al., Audiobox: Unified Audio Generation with Natural Language Prompts, arXiv:2312.15821, 2023.
>
> [2] A. Kumar, et al., TorchAudio-Squim: Reference-less Speech Quality and Intelligibility measures in TorchAudio, ICASSP, 2023.
> https://pytorch.org/audio/main/tutorials/squim_tutorial.html
>
> **Question 2:** Why is the PESQ score for the ground truth so low in Table 3?
>
> **Response:**
> The score for the ground truth (3.56) is low because we utilized the estimation method to make it comparable with the scores in the training dataset, as discussed above.
> Please see our response to Question 6 for the updated Table 3 with UTMOS (suggested criterion), PESQ and STOI scores obtained by reference audio.
>
> **Question 3:** Zero-shot TTS experiment was conducted by "continual" setting or "cross-utterance" setting?
>
> **Response:** The experiment was conducted by the continual setting. We have clarified this in Section 7.
>
> **Question 4:** Did the authors exclude the audio prompt portion of the synthesized audio when computing the speaker SIM?
>
> **Response:** Yes, we excluded the audio prompt when computing speaker SIM. This process was implemented as follows:
>
> ```
> with torch.no_grad():
>     emb1 = model(ref_wav[..., prompt_length:])
>     emb2 = model(gen_wav[..., prompt_length:])
> ```
>
> The table below compares speaker SIM values calculated with and without the audio prompt for HALL-E. Note that the increasing trend was observed with both VALL-E and HALL-E.
>
>
>
> | Prompt length        | VALL-E            | HALL-E | HALL-E (GT)
> |---------------|---------------|---------------|---------------|
> | 3s (default)            | 0.727 | 0.685 | 0.692 |
> | 10s                     | 0.801 | 0.750 | 0.778 |
> | 20s                     | 0.830 | 0.809 | 0.861 |
> | 40s                     | 0.851 | 0.846 | 0.930 |
> | 60s                     | 0.856 | 0.859 | 0.962 |

---

> > ### Author Response · Authors · 2024-11-23
> > **Reply to Reviewer aZfo (Part 2)**
> >
> > **Question 5:** Why is DNSMOS used? DNSMOS is specifically designed and trained to assess the cleanness of speech rather than its naturalness. I strongly suggest replacing it with a naturalness MOS metric, such as UTMOS.
> >
> > **Response:** We used DNSMOS following previous studies (e.g., [3, 4]) but we agree with the reviewer that UTMOS is more appropriate. We have added results of UTMOS to Tables 3, 4, 5, 6, 9 and 10. Below is a copy of Table 4. The results show that HALL-E exhibits significantly high UTMOS scores than VALL-E, indicating its strong performance in terms of naturalness. Thank you very much for your great advice.
> >
> > **Table 4.** Zero-shot TTS performance on MinutesSpeech test sets. Best results are marked in bold.
> >
> > | TTS model      | NAC model      | Training      | WER$\downarrow$       | SIM$\uparrow$       | WD$\downarrow$       | DNSMOS$\uparrow$              | UTMOS$\uparrow$              | QMOS$\uparrow$              | SMOS$\uparrow$              |
> > |------------------|----------------|----------------|---------------|------------|------------|-----------|----------------------|---------------------|---------------------|
> > | **test-90s**     |                |                |               |            |            |           |                      |                     |                     |                     |
> > | **GT**           | --             | --            | 10.30      | --         | --        | 3.79 $\pm$ 0.24      | 3.28$\pm$ 0.51     | 3.83 $\pm$ 0.30     |  --                   |
> > | **VALL-E**       | Encodec       | train-28s      | 39.77         | **0.726**  | 23.62      | 3.84 $\pm$ 0.19      | 3.61 $\pm$ 0.48     | 2.29 $\pm$ 0.32     | 2.04 $\pm$ 0.25     |
> > | **VALL-E†**      | Encodec       | train-90s      | 16.14         | 0.712      | **2.68**   | 3.87 $\pm$ 0.17      | 3.58 $\pm$ 0.60     | 2.48 $\pm$ 0.28     | 2.36 $\pm$ 0.26     |
> > | **HALL-E**       | MReQ-Encodec  | train-90s      | **9.79**      | 0.685      | 4.00       | **3.91 $\pm$ 0.21**  | **3.74 $\pm$ 0.37** | **3.35 $\pm$ 0.26** | **3.15 $\pm$ 0.26** |
> > | **test-180s**    |                |                |               |            |            |           |                      |                     |                     |                     |
> > | **GT**           | --             | --            | 10.20      | --         | --        | 3.78 $\pm$ 0.25      | 3.25 $\pm$ 0.50     | 4.26 $\pm$ 0.22     | --                   |
> > | **VALL-E**       | Encodec       | train-54s      | 36.52         | **0.706**  | 25.66      | 3.55 $\pm$ 0.53      | 3.15 $\pm$ 0.97     | 1.68 $\pm$ 0.25     | 1.70 $\pm$ 0.26     |
> > | **VALL-E†**      | Encodec       | train-180s     | 21.71         | 0.702      | 12.52      | 3.76 $\pm$ 0.30      | 3.55 $\pm$ 0.62     | 2.11 $\pm$ 0.26     | 2.15 $\pm$ 0.26     |
> > | **HALL-E**       | MReQ-Encodec  | train-180s     | **10.53**     | 0.660      | **5.79**   | **3.91 $\pm$ 0.19**  | **3.74 $\pm$ 0.37** | **3.38 $\pm$ 0.25** | **3.31 $\pm$ 0.23** |
> >
> > [3] X. Wang, et al., SpeechX: Neural Codec Language Model as a Versatile Speech Transformer, IEEE TASLP, vol. 32, pp. 1-14, 2024.
> >
> > [4] Sanyuan Chen, Shujie Liu, Long Zhou, Yanqing Liu, Xu Tan, Jinyu Li, Sheng Zhao, Yao Qian, and
> > Furu Wei. VALL-E 2: Neural codec language models are human parity zero-shot text to speech
> > synthesizers. arXiv preprint arXiv:2406.05370, 2024.

---

> > > ### Author Response · Authors · 2024-11-23
> > > **Reply to Reviewer aZfo (Part 3)**
> > >
> > > **Question 6:** Please present the speaker similarity and MOS scores (UTMOS?) for Table 3 and Table 10 (Table 9 in the revision).
> > >
> > > **Response:** Following the reviewer's suggestion, we have added speaker similarity (SIM) and UTMOS to the tables. Additionally, we analyzed the PESQ and STOI scores using reference audio in Table 3.
> > > As shown below, MReQ maintains performance in terms of WER, PESQ$^\dagger$, STOI, SIM and UTMOS. These results confirmed that MReQ effectively reduces the sampling rate while maintaining naturalness and clarity, which are essential for text-to-speech applications.
> > >
> > >
> > > **Table 3. Speech reconstruction on LibriSpeech.**
> > >
> > > | NAC (frame rate, Hz)         | WER$\downarrow$ | PESQ$\dagger$$\uparrow$ | PESQ$\uparrow$       | STOI$\uparrow$   | SIM$\uparrow$   | UTMOS$\uparrow$           |
> > > |------------------------------|------------------|---------------------------|------------------------|-------------------|------------------|----------------------------|
> > > | GT                           | 1.96            | 3.56                      | 4.64       | 1.00             | 1.00             | 4.05 $\pm$ 0.33            |
> > > | Encodec (8)                  | 100.0           | 1.25                      | 1.22 $\pm$ 0.08       | 0.32 $\pm$ 0.05   | 0.52             | 1.33 $\pm$ 0.00            |
> > > | Encodec (24)                 | 2.01            | 3.27                      | 3.68 $\pm$ 0.17       | 0.94 $\pm$ 0.03   | 0.90             | 3.82 $\pm$ 0.37            |
> > > | Encodec (48)                 | 2.00            | 3.45                      | 4.02 $\pm$ 0.13       | 0.96 $\pm$ 0.02   | 0.94             | 3.86 $\pm$ 0.36            |
> > > | Encodec (48) + MReQ          | 2.02            | 3.47                      | 3.89 $\pm$ 0.15       | 0.95 $\pm$ 0.02   | 0.92             | 3.89 $\pm$ 0.36            |
> > > | SpeechTokenizer (48)         | 2.00            | 3.52                      | 4.10 $\pm$ 0.11       | 0.96 $\pm$ 0.02   | 0.95             | 3.97 $\pm$ 0.33            |
> > > | SpeechTokenizer + MReQ (8,16,24,48) | 1.99     | 3.53                      | 3.96 $\pm$ 0.14       | 0.95 $\pm$ 0.02   | 0.93             | 4.01 $\pm$ 0.32            |
> > >
> > > $\dagger$ indicates PESQ (Kumar et al., 2023a).
> > >
> > >
> > > **Table 9. Ablation study for MReQ.**
> > >
> > > | Model            | WER$\downarrow$ | PESQ$\dagger$ $\uparrow$ | PESQ$\uparrow$         | UTMOS$\uparrow$       |
> > > |------------------|-----------------|----------------|--------------|-----------------------|
> > > | Proposed         | 2.02           | 3.47           | 3.89 $\pm$ 0.15 | 3.89 $\pm$ 0.36       |
> > > | w/o FLD loss     | 2.03           | 3.47           | 3.89 $\pm$ 0.15 | 3.87 $\pm$ 0.36       |
> > > | w/o HSR loss     | 2.15           | 3.44           | 3.79 $\pm$ 0.17 | 3.86 $\pm$ 0.37       |
> > > | w/o pre-training | 2.08           | 3.44           | 3.83 $\pm$ 0.17 | 3.85 $\pm$ 0.37       |
> > >
> > >
> > > **Question 7:** Please provide a breakdown of the AR and NAR model RTFs for Table 7.
> > >
> > > **Response:** We have replaced Table 7 with Figure 5 showing a breakdown, which confirms that the AR model is the primary computational bottleneck.
> > >
> > > We also have fixed errors and added the above-mentioned papers to the reference list. Thank you very much again for your careful review and valuable advice, which have improved the quality of our paper. We hope our responses have addressed your concerns.

---

> ### Comment · Reviewer_aZfo · 2024-11-23
> **Response to the authors' rebuttal**
>
> Thank you for the clarifications and revision. I think the authors addressed the major concerns that I had, and I raised the soundness score and overall rating of the paper. I still have a few clarification questions. I’d appreciate it if the authors could answer them.
> - QA2: I believe estimated PESQ score (PESQ with \dagger) is no longer necessary in Table 3 and Table 9. Please remove the column unless there’s a certain reason to keep them.
> - QA4: I have a few clarification questions.
>   - What is “HALL-E” and “HALL-E (GT)” in your explanation? Which is the version with and without the audio prompt?
>   - Why the number in the new table is different from the number in the prior version of the paper? For example, HALL-E score for 3s prompt was 0.663 in the original paper, but the result on the new table is 0.685 or 0.692, both does not match the number in the original paper.

---

> > ### Author Response · Authors · 2024-11-24
> >
> > Thank you very much for your quick reply and positive feedback.
> > > I believe estimated PESQ score (PESQ with \dagger) is no longer necessary in Table 3 and Table 9. Please remove the column unless there’s a certain reason to keep them.
> >
> > **Response:** We have removed the column. Thank you for your advice.
> >
> > > What is “HALL-E” and “HALL-E (GT)” in your explanation? Which is the version with and without the audio prompt?
> >
> > **Response:** HALL-E (GT) is the version with audio prompt.
> >
> > > Why the number in the new table is different from the number in the prior version of the paper? For example, HALL-E score for 3s prompt was 0.663 in the original paper, but the result on the new table is 0.685 or 0.692, both does not match the number in the original paper.
> >
> > **Response:** To report results with a prompt length of 60 seconds, the SIM scores were recalculated over a subset of audio samples with a minimum utterance length of 65 seconds (previously, this value was 25 seconds).

---

### Official Review · Reviewer_UZgt · 2024-10-31

**Soundness:** 3
**Presentation:** 3
**Contribution:** 3
**Rating:** 6
**Confidence:** 4

**Summary:**

To deal with the long-form speech synthesis problem, this paper propose a multi-resolution re-quantization method which hierarchically reorganizes discrete tokens with teacher-student distillations. Based on the hierarchical codec codes, an LLM-based TTS model HALL-E is proposed for speech synthesis. Besides these, a new benchmark dataset MinutesSpeech is introduced for minute-long speech synthesis.

This paper is well written and easy to follow.

**Strengths:**

1. This paper has three contributions 1). A multi-resolution requantization method MReQ is proposed to generate hierarchical codec codes. 2). A hierarchical LLM-base TTS model is proposed to predict MReQ codec codes. 3). A minute-long speech synthesis dataset is introduced.
2. This paper is well written and easy to follow.

**Weaknesses:**

1. Some important codec models are not compared with, such as DAC and Vocos.
2. MReQ is only tested on the speech reconstruction, without evaluated with audio dataset.
3. Some latest zero-shot TTS models are not compared with, such as Voicebox, E2 and VALL-E 2.

**Questions:**

1. Have you tried to train MReQ with a much smaller sampling rate, such as 2 or 4 Hz for the first layer?
2. What is the accuracy of the sub-encoder E to predict b_{k+1} based on a_{k+1}? Will this accuracy affect the final performance a lot?
3. For the L_{total}， are the weights of the three sub losses equal and all set to 1.0 ? Have you tried other settings?

---

> ### Author Response · Authors · 2024-11-24
> **Reply to Reviewer UZgt (Part 1)**
>
> Thank you very much for your positive feedback and constructive comments.
> Below, we respond to each question and comment.
>
> **Question 1:** Have you tried to train MReQ with a much smaller sampling rate, such as 2 or 4 Hz for the first layer?
>
> **Response:** Yes, Table 18 in the Appendix shows the results. Reducing the sampling rate below 8 Hz remains challenging. Since the audio token length is comparable to the text token length at a sampling rate of 8 Hz, further reductions would require new architectures and training algorithms. We leave further reduction of the token length for both audio and text streams for future work.
>
> **Question 2:** What is the accuracy of the sub-encoder E to predict $b_{k+1}$ based on $a_{k+1}$? Will this accuracy affect the final performance a lot?
>
> **Response:** The sub-encoder is frozen during the training of HALL-E. Because the role of this sub-module is to aggregate predicted $a_{k+1,1}, \cdots, a_{k+1,\alpha_{k+1}}$ into $b_{k+1}$, accuracy cannot be computed for this sub-module.
>
> **Question 3:**  For the $L_{\mathrm{total}}$, are the weights of the three sub losses equal and all set to 1.0 ? Have you tried other settings?
>
> **Response:** Appendix A.1 provides a more detailed description of $L_{\mathrm{total}}$, and Tables 11 and 12 list the weights for each loss function. The weights shown in Table 11 are from the official implementation of Encodec and have not been modified. Table 12 presents the weights for the FLD loss and HSR loss in our proposed method.
> These weights were selected after considering equal weighting and other weight values, choosing those that resulted in the greatest reduction in training loss.
>
> **Comment  1:** Some important codec models are not compared with, such as DAC and Vocos.
>
> **Response:** In preliminary experiments, we confirmed that SpeechTokenizer outperforms these conventional codec models. While training with various codec models could be interesting, it is not expected to improve the performance.
>
> **Comment  2:**  MReQ is only tested on the speech reconstruction, without evaluated with audio dataset.
>
> **Response:**
> Our method relies on the fact that speech retains semantic information primarily in relatively low-frequency components.
> By incorporating a hierarchical structure, MReQ achieved high reconstruction performance by preserving semantic information in low frame rate tokens while accurately retaining acoustic information in high frame rate tokens.
> Therefore, extending our method to general audio is non-trivial, and verifying this would require different large-scale downstream tasks, which is beyond the scope of this work.
> As the reviewer have pointed out, achieving a lower frame rate for general audio, not limited to speech, is an important direction for future research.

---

> > ### Author Response · Authors · 2024-11-24
> > **Reply to Reviewer UZgt (Part 2)**
> >
> > **Comment  3:** Some latest zero-shot TTS models are not compared with, such as Voicebox, E2 and VALL-E 2.
> >
> > **Response:**
> > We conducted evaluation experiments using MaskGCT [1] (a state-of-the-art NAR TTS) with the pre-trained model provided by the authors. Table 24 shows that the performance of MaskGCT significantly decreases as the length of synthesized speech increases.
> > This indicates that even with NAR TTS, generating long speech in a single inference procedure is challenging.
> > In the 30-second test setting in Table 25, the WER of MaskGCT is lower than VALL-E and HALL-E. However, its UTMOS is worse compared to HALL-E.
> > In the 90-second test setting in Table 26, only HALL-E achieves a WER lower than 10.0%,
> > with the highest UTMOS.
> > For generating spontaneous speech, producing more natural durations becomes crucial.
> > AR models excel in this aspect compared to NAR models, which likely contributes to the higher UTMOS scores observed.
> > We have added these results in Appendix E.
> >
> > The comparison suggested by the reviewer provided a clearer demonstration of the proposed method's effectiveness and relevance. We greatly appreciate the reviewer's insightful suggestion.
> >
> > [1] Y. Wang, et al., MaskGCT: Zero-Shot Text-to-Speech with Masked Generative Codec Transformer, arXiv:2409.00750.
> >
> > **Table 24. MaskGCT performance by test duration**
> >
> > | Method    | Duration | WER   | SIM   | DNSMOS       | UTMOS            |
> > |-----------|---------------|-------|-------|--------------|------------------|
> > | MaskGCT   | 30 sec.           | 7.74  | 0.763 | 3.95 $\pm$ 0.23  | 3.42 $\pm$ 0.46    |
> > | MaskGCT   | 45 sec.           | 10.75 | 0.763 | 3.94 $\pm$ 0.22  | 3.39 $\pm$ 0.49    |
> > | MaskGCT   | 60 sec.           | 23.08 | 0.757 | 3.92 $\pm$ 0.23  | 3.30 $\pm$ 0.54    |
> > | MaskGCT   | 90 sec.           | 51.12 | 0.691 | 3.57 $\pm$ 0.51  | 2.57 $\pm$ 1.02    |
> >
> > **Table 25. Performance comparison in 30-second test**
> >
> > | Method        | Duration | WER   | SIM   | DNSMOS       | UTMOS            |
> > |---------------|---------------|-------|-------|--------------|------------------|
> > | GT            | 30 sec.           | 10.65 | -     | 3.74 $\pm$ 0.29            | 3.20 $\pm$ 0.51               |
> > | VALL-E train-28s    | 30 sec.           | 11.84 | 0.731 | 3.84 $\pm$ 0.19        | 3.61 $\pm$ 0.41    |
> > | VALL-E train-90s    | 30 sec.           | 14.83 | 0.695 | 3.82 $\pm$ 0.20        | 3.46 $\pm$ 0.50    |
> > | HALL-E train-90s    | 30 sec.           | 10.88 | 0.685 | 3.87 $\pm$ 0.21        | 3.61 $\pm$ 0.38    |
> > | MaskGCT       | 30 sec.           | 7.74  | 0.763 | 3.95 $\pm$ 0.23  | 3.42 $\pm$ 0.46    |
> >
> > **Table 26. Performance comparison in 90-second test**
> >
> > | Method        | Duration | WER   | SIM   | DNSMOS       | UTMOS            |
> > |---------------|---------------|-------|-------|--------------|------------------|
> > | GT            | 90 sec.           | 10.30  | -     | 3.79 $\pm$ 0.24  | 3.28 $\pm$ 0.51      |
> > | VALL-E train-28s    | 90 sec.           | 39.77 | 0.726 | 3.84 $\pm$ 0.19  | 3.61 $\pm$ 0.48      |
> > | VALL-E train-90s    | 90 sec.           | 16.14 | 0.712 | 3.87 $\pm$ 0.17  | 3.58 $\pm$ 0.60      |
> > | HALL-E train-90s    | 90 sec.           | 9.79  | 0.685 | 3.91 $\pm$ 0.21  | 3.74 $\pm$ 0.37      |
> > | MaskGCT       | 90 sec.           | 51.12 | 0.691 | 3.57 $\pm$ 0.51  | 2.57 $\pm$ 1.02    |

---

> > > ### Comment · Reviewer_UZgt · 2024-11-27
> > >
> > > Thanks the author for the reply. I keep my original score, reflecting the contribution of this paper.

---

### Official Review · Reviewer_1NPx · 2024-11-03

**Soundness:** 3
**Presentation:** 3
**Contribution:** 4
**Rating:** 6
**Confidence:** 4

**Summary:**

This paper presents two post-training approaches to resolve the long context length issues for modern transformer-based autoregressive TTS models based on NAC models. MReQ as a framework reduces the framerate by introducing a novel Multi-resolution vector quantization module that allows the user to decompose an existing NAC model into several different code books, each of which operate at a different framerate. This allows them to reduce the bottleneck of the AR component of LLM-based TTS by reducing the frequency of the first component down to 8 Hz. To train this MRVQ module, student-teacher distillation is required. Using this module, this work presents HALL-E, a hierarchical TTS model that generates tokens based on the MReQ tokens. This paper also introduces MinutesSpeech, a benchmark dataset for TTS synthesis that is curated for long-form speech.

**Strengths:**

Idea is pretty interesting and novel and well motivated. The MReQ module is fairly complex, but diagrams and descriptions of the technique are well written. Manuscript is very detailed and training and post-training are well documented. Along with the code release, results look to be easily replicable.

Authors go through great lengths to train and evaluate everything from scratch and provide extensive results for reconstruction and zero-shot TTS which included both automatic and subjective results. Experiments were run on multiple NAC models, showing that this technique can generalize to pre-existing NAC models.

Benchmark dataset is a welcome addition and the generation and evaluation of the dataset are well documented. Results show RTF improvements, indicating that this technique indeed reduces the bottleneck behind transformer-based TTS.

**Weaknesses:**

* One weakness in the study is that authors do not consider distribution shifts between the different dataset used to train the NAC model and the dataset used for post training. In their experiments, they pretrain their own NAC models (Encodec and Speechtokenizer) on the MinutesSpeech training set and did post training with MReQ with the same training data. However, in practice, researchers will not have access to the datasets used to pretrain these NAC models, so there may be some distribution shift. A similar issue occurs with quantization, where you need some training data to calibrate the model. Additionally, while SpeechTokenizer is meant to tokenize speech, Encodec was originally designed to encode speech along with other kinds of audio. It is unclear whether this capability will be preserved after post-training. Ablation studies do not cover what happens when you try to use this technique with an off-the shelf NAC model like SpeechTokenizer or Encodec off the shelf.

* The statement on line 211/212 that Training NAC models with the MRVQ module does not seem to be justified or elaborated on. Is this just saying that training this kind of codec model from scratch rather than from a pretrained model is difficult? It seems like from Table 10, that for the MReQ model, starting this kind of training w/o pretraining seems to not affect the WER or RESQ significantly.

* The qualitative results in the ablation study near line 495 are not well explained. The statement that this waveform is unnatural with almost no silence is not clear without a transcription. From a glance, this seems to be a problem with the prosody of the generation, but this is unclear without hearing the samples. Appendix D1 also does not seem to elaborate on this much. I tried to listen to the samples from MinutesSpeech test-90s, and it seems like this is primarily a prosody issue as the Valle model seems to speak at a very consistent rate with much louder inhalation sounds.

* Furthermore, the ablation study for “Can HALL-E handle long audio prompts?” seems trivial, it should be expected that longer voice prompts gives the model more context on the user’s speech. It would be more interesting to see if this effect exists or is less pronounced in the VALLE model.

* There are also some wording errors in the manuscript. For instance, in the implementation details on page 4, lines 199 to 201, the manuscript says that “Figure 2a shows the MRVQ module applied to the Encodec model”, however, Figure 2a actually just shows the normal Encodec model as a teacher model. This should actually be Figure 2b that shows the MRVQ module applied to the Encodec model. Furthermore, Line 201 states that “Figure 2b shows the LRVQ block”, however, it is actually Figure 2c that shows the LRVQ block.  Also, should the statement in line 485/486: “The results indicate that both losses and post-training play important roles in the overall performance.” say that both losses and “pre-training” play important roles, since the second row of Table 10 is about the pre-training of the Encodec model?

**Questions:**

* This technique seems well suited for LLM-based TTS solutions, similar to Valle with AR and NAR components. However, can you also say anything about how this kind of encoding might be used as a general purpose codec for speech in general? It would be interesting if this model could have lower bitrates or higher robustness. Furthermore, does this means that the PreQ and PostQ components of your LRVQ blocks would be unnecessary since they are primarily used to train the NAR components of the TTS model?

* With a VALLE model based on Encodec, once can remove RVQ code sequences from the NAR generation to get faster generation, in exchange for coarser generation. For this MReQ based paradigm, how does the audio quality here change?

* For the second ablation in section 7.4, how does this effect appear with the VALLE model that you trained?

* For the third ablation in section 7.4, it is very surprising that even w/o pretraining, it does not seem like this technique suffers significantly (compared to the performance drop in Table 11 for HALL-E). Why is that?

* The splits of the dataset in Table 2 and the description of why these splits were created in Section 7.1 seems very arbitrary and over designed for these experiments. Are these training splits deduplicated from each other?

---

> ### Author Response · Authors · 2024-11-23
> **Reply to Reviewer 1NPx (Part 1)**
>
> Thank you very much for your insightful review and positive feedback. Below, we respond to each question and comment.
>
> **Question 1.1:** This technique seems well suited for LLM-based TTS solutions, similar to VALL-E with AR and NAR components. However, can you also say anything about how this kind of encoding might be used as a general purpose codec for speech in general?
>
> **Response:** The proposed approach could be advantageous for generative speech tasks, such as speech-to-speech translation beyond TTS, due to its ability to model long-term context using low sampling rate tokens.
> To reduce the bitrate, it would be preferable to improve the teacher model rather than rely on the distillation framework, as shown in Table 19.
>
> **Question 1.2:** Furthermore, does this means that the PreQ and PostQ components of your LRVQ blocks would be unnecessary since they are primarily used to train the NAR components of the TTS model?
>
> **Response:** It depends on the task and downstream architecture. For discriminative tasks, including automatic speech recognition, if learning a discriminator over the features of the first quantization layer is sufficient, the PreQ and PostQ components may no longer be needed.
> For generative architectures that do not rely on NAR components, such as Fisher and Moshi using the AR+AR combination, components like PreQ and PostQ may be helpful but this requires further investigation.
>
> **Question 2:** With a VALLE model based on Encodec, once can remove RVQ code sequences from the NAR generation to get faster generation, in exchange for coarser generation. For this MReQ based paradigm, how does the audio quality here change?
>
> **Response:** If our understanding is correct, the reviewer's intent is that using only the output of the AR model and omitting the NAR model can result in faster generation, albeit with coarser quality.
>
> The table below shows the results when audio is generated using only the AR model, omitting the NAR model in each method.
> As expected, the overall performance deteriorates; however, trends in metrics such as WER remain similar to those observed in Table 4, indicating that our method (HALL-E) continues to demonstrate effectiveness.
> Nevertheless, as illustrated in Figure 5, the NAR model does not demand significant computational resources, so it is advisable to employ it to improve quality.
>
> | Method        | WER   | SIM   | DNSMOS       | UTMOS            |
> |---------------|-------|-------|--------------|------------------|
> | VALL-E train-28s    | 42.62 | 0.616 | 2.878 $\pm$ 0.143  | 1.449 $\pm$   0.182  |
> | VALL-E train-90s    | 19.34 | 0.624 | 2.860 $\pm$  0.118 | 1.434 $\pm$   0.173  |
> | HALL-E train-90s    | 16.23 | 0.616 | 3.003 $\pm$ 0.194  | 1.343 $\pm$   0.112  |
>
>
>
> **Question 3:** For the second ablation in section 7.4, how does this effect appear with the VALLE model that you trained?
>
> **Response:** In response to the feedback, we have added results for VALL-E (trained on MinutesSpeech-90s) to Table 8, where the audio prompt length was varied from 3 seconds to 60 seconds. The SIM scores are recalculated over audio samples with a minimum utterance length of 65 seconds.
>
> As shown below, a similar effect was observed with VALL-E. However, the gap between HALL-E and VALL-E narrowed as the prompt length increased. Finally, HALL-E slightly outperformed VALL-E at a prompt length of 60 seconds, demonstrating that highly accurate cloning is achievable with HALL-E.
>
> **Table 8. SIM as a function of audio prompt length.**
>
> | Prompt length        | VALL-E            | HALL-E |
> |---------------|---------------|---------------|
> | 3s (default)            | 0.727 | 0.685 |
> | 10s                     | 0.801 | 0.750 |
> | 20s                     | 0.830 | 0.809 |
> | 40s                     | 0.851 | 0.846 |
> | 60s                     | 0.856 | 0.859 |
>
> **Question 4:** For the third ablation in section 7.4, it is very surprising that even w/o pretraining, it does not seem like this technique suffers significantly (compared to the performance drop in Table 11 for HALL-E). Why is that?
>
> **Response:** This is because the MRVQ module involves three 48 Hz quantization layers. It is known that with Encodec, reducing the number of RVQ layers to 6 or 3 degrades performance but does not cause collapsed speech. Therefore, the performance drop was not significant compared to that of HALL-E.

---

> > ### Author Response · Authors · 2024-11-23
> > **Reply to Reviewer 1NPx (Part 2)**
> >
> > **Question 5:** The splits of the dataset in Table 2 and the description of why these splits were created in Section 7.1 seems very arbitrary and over designed for these experiments. Are these training splits deduplicated from each other?
> >
> > **Response:** The training sets share the same transcription and audio data. The difference between these sets lies in the audio lengths when segmenting long original audio files.
> > Our initial design focused on only two durations: 90 seconds and 180 seconds. However, we decided to add 28 seconds and 54 seconds for a fair comparison with VALL-E.
> > As shown in Table 2, the train-28s set was created to have the same token length as train-90s in the "Sum" column.
> > This aligns the input length of the AR language models for HALL-E and VALL-E.
> > Similarly, train-54s was created to align with train-180s.
> >
> > **Comment 1:** In practice, researchers will not have access to the datasets used to pretrain these NAC models, so there may be some distribution shift.
> >
> > **Response:** Investigating distribution shift using models pretrained on inaccessible data sounds interesting; however, it also limits experimental opportunities. For instance, when comparing NAC architectures (e.g., Encodec and SpeechTokenizer) in our framework, it would be necessary to use the same data for pretraining to ensure a fair comparison. Without access to the pretraining data, it is also challenging to discuss the extent of a distribution shift. We leave dataset curation for non-speech audio data as future work.
> >
> > **Comment 2:** It seems like from Table 10, that for the MReQ model, starting this kind of training w/o pretraining seems to not affect the WER or RESQ significantly.
> >
> > **Response:** As discussed in Question 4, this is because the MRVQ module involves high frame rate layers.
> >
> > **Comment 3:** The qualitative results in the ablation study near line 495 are not well explained. The statement that this waveform is unnatural with almost no silence is not clear without a transcription.
> >
> > **Response:** Our intent behind the sentence was to convey that the AR model of VALL-E struggles with duration prediction. Following reviewer's suggestion, we manually annotated the segments for each word to show the transcription in Figure 6 (Section 7.4) and in Figure 8 (Appendix D.1). As shown in the first sentence highlighted in red, the duration of the proper noun "Cam Ross" differs significantly across the models. In the speech generated by HALL-E, this phrase is pronounced more slowly and clearly, closely resembling the ground truth.
> >
> > **Comment 4.** Furthermore, the ablation study for "Can HALL-E handle long audio prompts?" seems trivial, it should be expected that longer voice prompts gives the model more context on the user's speech. It would be more interesting to see if this effect exists or is less pronounced in the VALLE model.
> >
> > **Response:** As discussed in Question 3, we have added results of VALL-E. The effect exists for both HALL-E and VALL-E. With a prompt length of 60 seconds, HALL-E outperformed VALL-E. Since the reduced frame rate achieved by MReQ enables efficient generation even with long prompts, the result demonstrates that HALL-E is capable of achieving high-fidelity synthesis, highlighting its practical value.
> >
> > We also have corrected wording errors pointed out by the reviewer. Thank you very much for your careful review and great advice, which have improved the quality of our paper.

---

> > > ### Comment · Reviewer_1NPx · 2024-11-26
> > >
> > > Thank you, authors, for your reply. These questions were solely for clarification purposes and for the improvement of your manuscript. I maintain my original score, which already reflects the contribution of this work.

---

### Official Review · Reviewer_Fwrz · 2024-11-04

**Soundness:** 2
**Presentation:** 1
**Contribution:** 3
**Rating:** 6
**Confidence:** 3

**Summary:**

The paper presents HALL-E, a neural codec language model aimed at addressing challenges in minute-long zero-shot text-to-speech (TTS) synthesis. It introduces Multi-Resolution Requantization (MReQ) to reduce frame rates in neural audio codec (NAC) models, and proposes HALL-E, which leverages MReQ for efficient and effective long-form speech synthesis. The work also introduces MinutesSpeech, a new speech dataset for long-context conversational TTS. The experimental results are extensive, showing that the proposed method can outperform baseline models both quantitatively and qualitatively.

**Strengths:**

* The paper offers thorough quantitative and qualitative results, demonstrating that the proposed generative modeling outperforms baseline methods in long-form speech synthesis.
* The introduction of MinutesSpeech is a noteworthy addition to TTS research, providing high-quality, long-context speech data. The illustrated curation and preprocessing techniques further enhance its utility.
* The ablation studies on MRVQ provide insights into the necessity of each component, justifying their inclusion in the architecture.

**Weaknesses:**

* Although the MReQ module achieves low-resolution quantization while maintaining sample quality, its design is excessively complicated, posing challenges for future research to build upon. The LRVQ design incorporates multiple RVQ modules—pre-quantizer, main quantizer, and post-quantizer—alongside the additional HSR loss, making it significantly more complex than prior RVQ methods. Furthermore, the use of multiple RVQ quantizations within the NAR transformer of HALL-E adds another layer of complexity, reducing the accessibility of the proposed approach.
* The paper only compares the proposed method with VALL-E and does not include recent zero-shot TTS baselines. Considering that non-autoregressive modeling can be advantageous for long speech generation, incorporating comparisons with state-of-the-art non-autoregressive zero-shot TTS models would better demonstrate the effectiveness and relevance of the proposed method.

**Questions:**

* The description of the LRVQ modules appears to omit the use of commitment loss for residual vector quantization. If the LRVQ module does not require commitment loss, the authors would explain the reasoning behind this and how the training is stabilized without it.
* Providing a detailed description of the pre-trained LLM-based TTS model that HALL-E is post-trained on would greatly aid in understanding the training process and setup of the proposed model.

---

> ### Author Response · Authors · 2024-11-23
>
> We appreciate your positive feedback and detailed comments. Below, we respond to each question and comment.
>
> **Question 1:** The description of the LRVQ modules appears to omit the use of commitment loss for residual vector quantization.
>
> **Response:** The commitment loss was not omitted and is included in the loss term $L_{NAC}$. For clarification, we have added the detailed loss definition to the appendix A.1.
>
> **Question 2:** Providing a detailed description of the pre-trained LLM-based TTS model.
>
> **Response:** Pre-training was performed without incorporating the sub-modules of MRVQ. Thus, the pre-trained model is identical to VALL-E. We have clarified this in Section 7.1.
>
> **Comment 1:** Although the MReQ module achieves low-resolution quantization while maintaining sample quality, its design is excessively complicated, posing challenges for future research to build upon.
>
> **Response:** While the proposed architecture for achieving high zero-shot TTS performance may appear complicated, the core idea to introduce a recursive architecture to RVQ is simple. In particular, our implementation of MReQ in audiocraft/models/encoder.py within the provided code is not complicated, and is expected to facilitate future research and development.
>
> **Comment 2:** Comparisons with state-of-the-art non-autoregressive zero-shot TTS models would better demonstrate the effectiveness and relevance of the proposed method.
>
> **Response:**
> We conducted evaluation experiments using MaskGCT [1] (a state-of-the-art NAR TTS) with the pre-trained model provided by the authors.
>
> Table 24 shows that the performance of MaskGCT significantly decreases as the length of synthesized speech increases.
> This indicates that even with NAR TTS, generating long speech in a single inference procedure is challenging.
> In the 30-second test setting in Table 25, the WER of MaskGCT is lower than VALL-E and HALL-E. However, its UTMOS is worse compared to HALL-E.
> In the 90-second test setting in Table 26, only HALL-E achieves a WER lower than 10.0%,
> with the highest UTMOS.
> For generating spontaneous speech, producing more natural durations becomes crucial.
> AR models excel in this aspect compared to NAR models, which likely contributes to the higher UTMOS scores observed.
> We have added these results in Appendix E.
>
> The comparison suggested by the reviewer provided a clearer demonstration of the proposed method's effectiveness and relevance. We greatly appreciate the reviewer's insightful suggestion.
>
> [1] Y. Wang, et al., MaskGCT: Zero-Shot Text-to-Speech with Masked Generative Codec Transformer, arXiv:2409.00750.
>
> **Table 24. MaskGCT performance by test duration**
>
> | Method    | Duration | WER   | SIM   | DNSMOS       | UTMOS            |
> |-----------|---------------|-------|-------|--------------|------------------|
> | MaskGCT   | 30 sec.           | 7.74  | 0.763 | 3.95 $\pm$ 0.23  | 3.42 $\pm$ 0.46    |
> | MaskGCT   | 45 sec.           | 10.75 | 0.763 | 3.94 $\pm$ 0.22  | 3.39 $\pm$ 0.49    |
> | MaskGCT   | 60 sec.           | 23.08 | 0.757 | 3.92 $\pm$ 0.23  | 3.30 $\pm$ 0.54    |
> | MaskGCT   | 90 sec.           | 51.12 | 0.691 | 3.57 $\pm$ 0.51  | 2.57 $\pm$ 1.02    |
>
> **Table 25. Performance comparison in 30-second test**
>
> | Method        | Duration | WER   | SIM   | DNSMOS       | UTMOS            |
> |---------------|---------------|-------|-------|--------------|------------------|
> | GT            | 30 sec.           | 10.65 | -     | 3.74 $\pm$ 0.29            | 3.20 $\pm$ 0.51               |
> | VALL-E train-28s    | 30 sec.           | 11.84 | 0.731 | 3.84 $\pm$ 0.19        | 3.61 $\pm$ 0.41    |
> | VALL-E train-90s    | 30 sec.           | 14.83 | 0.695 | 3.82 $\pm$ 0.20        | 3.46 $\pm$ 0.50    |
> | HALL-E train-90s    | 30 sec.           | 10.88 | 0.685 | 3.87 $\pm$ 0.21        | 3.61 $\pm$ 0.38    |
> | MaskGCT       | 30 sec.           | 7.74  | 0.763 | 3.95 $\pm$ 0.23  | 3.42 $\pm$ 0.46    |
>
> **Table 26. Performance comparison in 90-second test**
>
> | Method        | Duration | WER   | SIM   | DNSMOS       | UTMOS            |
> |---------------|---------------|-------|-------|--------------|------------------|
> | GT            | 90 sec.           | 10.30  | -     | 3.79 $\pm$ 0.24  | 3.28 $\pm$ 0.51      |
> | VALL-E train-28s    | 90 sec.           | 39.77 | 0.726 | 3.84 $\pm$ 0.19  | 3.61 $\pm$ 0.48      |
> | VALL-E train-90s    | 90 sec.           | 16.14 | 0.712 | 3.87 $\pm$ 0.17  | 3.58 $\pm$ 0.60      |
> | HALL-E train-90s    | 90 sec.           | 9.79  | 0.685 | 3.91 $\pm$ 0.21  | 3.74 $\pm$ 0.37      |
> | MaskGCT       | 90 sec.           | 51.12 | 0.691 | 3.57 $\pm$ 0.51  | 2.57 $\pm$ 1.02    |

---

> > ### Comment · Reviewer_Fwrz · 2024-11-25
> >
> > I appreciate the authors’ efforts in addressing my questions and concerns to some extent. The clarification regarding the NAC loss and the pretrained TTS model is helpful. While I still find the modeling approach to be quite complex, even compared to certain modules in audiocraft/models/encodec.py, I appreciate that the authors have provided the implementation of this work including the encodec file. Furthermore, there remains some uncertainty about how the proposed model outperforms recent state-of-the-art NAR methods or how the NAR TTS model, MaskGCT, might perform if trained on a minutes-long TTS dataset. However, the newly conducted experiment still offers meaningful insights, demonstrating that autoregressive models, when combined with the proposed method, can generate coherent long-form speech. I thank the authors for their efforts to respond, and I remain confident in my assessment.

---

### Public Comment · ~Ewald_Enzinger1 · 2025-02-21
**Issues running tts_demo.py script**

Dear authors,

I very much enjoyed reading your paper! Thank you for releasing the source code as supplementary material.
I tried to run the tts_demo.py script, but ran into some issues:

It appears the SpeechGenSolver class implementation is missing from audiocraft/solvers/valle_ar.py:
```
(halle) ~/HALL-E/scripts $ python tts_demo.py
Traceback (most recent call last):
  File "/home/user/HALL-E/scripts/tts_demo.py", line 6, in <module>
    from audiocraft.models import (
  File "/home/user/HALL-E/audiocraft/__init__.py", line 24, in <module>
    from . import data, models, modules
  File "/home/user/HALL-E/audiocraft/models/__init__.py", line 15, in <module>
    from .halle import Halle
  File "/home/user/HALL-E/audiocraft/models/halle.py", line 20, in <module>
    from ..solvers.valle_ar import SpeechGenSolver
  File "/home/user/HALL-E/audiocraft/solvers/__init__.py", line 21, in <module>
    from .halle_ar import HalleARSolver
  File "/home/user/HALL-E/audiocraft/solvers/halle_ar.py", line 35, in <module>
    from .valle_ar import SpeechGenSolver
ImportError: cannot import name 'SpeechGenSolver' from 'audiocraft.solvers.valle_ar' (/home/user/HALL-E/audiocraft/solvers/valle_ar.py)
```
With the assumption that ValleARSolver is equivalent to SpeechGenSolver, I tried to fix the imports, but ran into another issue: The function "load_hier_lm_model" is missing from audiocraft/models/loaders.py:
```
$ python tts_demo.py
Traceback (most recent call last):
  File "/home/user/HALL-E/scripts/tts_demo.py", line 6, in <module>
    from audiocraft.models import (
  File "/home/user/HALL-E/audiocraft/__init__.py", line 24, in <module>
    from . import data, models, modules
  File "/home/user/HALL-E/audiocraft/models/__init__.py", line 16, in <module>
    from .halle import Halle
  File "/home/user/HALL-E/audiocraft/models/halle.py", line 25, in <module>
    from .loaders import load_compression_model, load_hier_lm_model
ImportError: cannot import name 'load_hier_lm_model' from 'audiocraft.models.loaders' (/home/user/HALL-E/audiocraft/models/loaders.py)
```
Are you planning to release an updated source code repository?

Thanks,
Ewald

---

### Public Comment · ~Yuancheng_Wang1 · 2025-03-18
**Repo Github not Found**

Dear authors,

Thanks for your great work! However, it seems like the repo github https://yutonishimura-v2.github.io/HALL-E_DEMO can not found. And I would like to ask if you are going to open source the MinutesSpeech dataset?

Thanks, Yuancheng

---

### Meta-Review · Area_Chair_C9KC · 2024-12-20

**Metareview:**

In this work, the authors targeted to resolve the challenge of long-form TTS. The high frame rate results in the long length of audio tokens, which makes it different for autoregressive language models to generate tokens. Representative works such as VALL-E encounter challenges in generating high-quality audio for extended sequences. Consequently, this research is significant for advancing speech generation beyond 10+ seconds.

There are two main contributions of this paper: 1) Multi-Resolution Requantization (MReQ) which is used to reduce the frame rate of neural codec. 2) HALL-E which is an LLM-based TTS model to predict hierarchical tokens of MReQ. Furthermore, the authors have developed a new dataset comprising 40,000 hours of training data, which will support the community in researching long-form speech generation.

The work is sufficiently novel to address a very meaningful problem of speech generation. The experiments are well-executed with convincing results.  During the rebuttal, the authors actively answered reviewers’ questions with clarified details. Therefore, the work is worth publication in ICLR.

In summary, the strength of this paper is effectively addressing the challenging problem of synthesizing speech longer than 10 seconds by utilizing the proposed MReQ to reduce the frame rate and HALL-E to model the hierarchical tokens of MReQ. However, a noted weakness is the lack of comparison with recent advanced zero-shot TTS models such as VALL-E 2, which has shown significant improvement over VALL-E used in the paper, despite the inclusion of a comparison with MaskGCT during the rebuttal.

**Additional Comments On Reviewer Discussion:**

The initial scores of this paper are consistent with 5, 6, 6, 6, 6. The final scores for this paper are 6, 6, 6, 6, and 8. The authors did a very good job during rebuttal to dismiss the reviewers’ concerns.
Specifically,  Reviewer aZfo originally raised lots of concerns on the experience setup such as unreasonable usage of PESQ and why DNSMOS is used. The authors dismissed all those concerns item by item. As a result, the reviewer raised the score to 8.
To respond to the review comments of comparing to SOTA models, the authors added experiments of MaskGCT.

---

### Decision · Program_Chairs · 2025-01-22

Accept (Poster)